# Genetic Responses of Metabolically Active *Limnospira indica* Strain PCC 8005 Exposed to γ-Radiation during Its Lifecycle

**DOI:** 10.3390/microorganisms9081626

**Published:** 2021-07-30

**Authors:** Anu Yadav, Laurens Maertens, Tim Meese, Filip Van Nieuwerburgh, Mohamed Mysara, Natalie Leys, Ann Cuypers, Paul Jaak Janssen

**Affiliations:** 1Interdisciplinary Biosciences, Microbiology Unit, Belgian Nuclear Research Centre (SCKCEN), 2400 Mol, Belgium; anuiisermohali@gmail.com (A.Y.); laurens.maertens@sckcen.be (L.M.); mohamed.mysara.ahmed@sckcen.be (M.M.); natalie.leys@sckcen.be (N.L.); 2Environmental Biology, Centre for Environmental Sciences, Hasselt University, 3590 Diepenbeek, Belgium; ann.cuypers@uhasselt.be; 3Research Unit in Biology of Microorganisms (URBM), Narilis Institute, University of Namur, 5000 Namur, Belgium; 4Laboratory of Pharmaceutical Biotechnology, Ghent University, 9000 Ghent, Belgium; tim.meese@ugent.be (T.M.); filip.vannieuwerburgh@ugent.be (F.V.N.)

**Keywords:** *Limnospira*, *Arthrospira*, gamma radiation, expression analysis, RNA-Seq, radiation resistance, morphology, genomics, genetic response

## Abstract

Two morphotypes of the cyanobacterial *Limnospira* *indica* (formerly *Arthrospira* sp.) strain PCC 8005, denoted as P2 (straight trichomes) and P6 (helical trichomes), were subjected to chronic gamma radiation from spent nuclear fuel (SNF) rods at a dose rate of ca. 80 Gy·h^−1^ for one mass doubling period (approximately 3 days) under continuous light with photoautotrophic metabolism fully active. Samples were taken for post-irradiation growth recovery and RNA-Seq transcriptional analysis at time intervals of 15, 40, and 71.5 h corresponding to cumulative doses of ca. 1450, 3200, and 5700 Gy, respectively. Both morphotypes, which were previously reported by us to display different antioxidant capacities and differ at the genomic level in 168 SNPs, 48 indels and 4 large insertions, recovered equally well from 1450 and 3200 Gy. However, while the P2 straight type recovered from 5700 Gy by regaining normal growth within 6 days, the P6 helical type took about 13 days to recover from this dose, indicating differences in their radiation tolerance and response. To investigate these differences, P2 and P6 cells exposed to the intermediate dose of gamma radiation (3200 Gy) were analyzed for differential gene expression by RNA-Seq analysis. Prior to batch normalization, a total of 1553 genes (887 and 666 of P2 and P6, respectively, with 352 genes in common) were selected based on a two-fold change in expression and a false discovery rate FDR smaller or equal to 0.05. About 85% of these 1553 genes encoded products of yet unknown function. Of the 229 remaining genes, 171 had a defined function while 58 genes were transcribed into non-coding RNA including 21 tRNAs (all downregulated). Batch normalization resulted in 660 differentially expressed genes with 98 having a function and 32 encoding RNA. From PCC 8005-P2 and PCC 8005-P6 expression patterns, it emerges that although the cellular routes used by the two substrains to cope with ionizing radiation do overlap to a large extent, both strains displayed a distinct preference of priorities.

## 1. Introduction

Due to the large-scale industrial production of the cyanobacterium *Limnospira* with its high nutritive value as a feed and food supplement and its use as a major cell factory for a range of biopharmaceuticals and added-value chemical compounds, a thorough understanding of the various genetic and cellular mechanisms in response to variable environmental parameters is important. Hence, the behavior of *Limnospira* under different environmental conditions has been studied by whole-genome transcriptomic analysis including nitrogen deprivation [1,2], elevated temperature [3], and sulfate deficiency [4]. These transcriptomic analyses were enabled by concurrent genome sequencing efforts across the globe, with the genomic sequences of at least seven strains now available [5].

About three decades ago the cyanobacterial *Arthrospira* sp. strain PCC 8005 was chosen by the European Space Agency as a principal organism in the Micro-Ecological Life Support System Alternative (MELiSSA) (https://www.melissafoundation.org/*)* for efficient O_2_ production and recycling of CO_2_, and the production of biomass as a highly nutritional end product [6]. It recently was given the status of type strain to the newly proposed species *Limnospira indica* [7]. The strain’s genome was fully sequenced by us [8,9] and annotated using the MicroScope/MaGe platform [10] rendering an assembly of six ordered contigs spanning together 6,228,153 bp and holding the genetic information for 6345 coding regions (CDS) and 337 genes transcribed in non-coding RNA (ncRNA) (currently known as ARTHROv5—updated version 5 of 15 February 2014, available at NCBI under GenBank assembly accession number GCA_000973065.1; also available from Appendix A or from the MicroScope/MaGe platform [10] upon simple request to the corresponding author for conditional access). During our subsequent studies, we found that strain PCC 8005 was tolerant to extremely high doses of gamma rays withstanding cumulative radiation doses of up to 5000 Gy, albeit with a delayed recovery in growth [11,12]. From this earlier work, it became clear that *L. indica* PCC 8005 deploys a cascade of modes in its response to high doses of gamma radiation: an emergency mode in which cells quickly try to adapt to the sudden radiation stress by shutting down central processes such as photosynthesis, carbon fixation, and nitrogen assimilation, a survival mode redirecting the freed-up cellular resources towards detoxification, protein protection, and DNA repair, and a recovery mode in which vital pathways for energy maintenance and metabolic activity are gradually restarted. The results of Badri et al. [11,12] also suggested that *L. indica* PCC 8005 may not primarily rely on enzymatic systems to overcome oxidative stress incited by ionizing radiation (IR) (i.e., through the action of so-called reactive oxygen species or ROS) but rather that non-enzymatic systems are at play, and that compounds such glutathione and other short aromatic peptides, lycopene, β-carotene, α-tocopherol, and Mn^2+^-complexes have a critical role in *Limnospira* IR resistance which is likely achieved by a “metabolic route” deploying a combination of highly coordinated physiological processes. In a more recent study, we observed an irreversible morphological change in PCC 8005 subcultures, i.e., from only helical to only straight trichomes; these morphotypes displayed differences in growth rate, buoyancy, and resistance to gamma radiation [13]. We also found marked differences between these subtypes in antioxidant capacity, pigment content, and trehalose concentration, while whole-genome comparison revealed a difference of 168 SNPs, 48 indels and four large insertions affecting 41 coding regions across both genomes [13].

The doubling time of *L. indica* PCC 8005 is about 3 days and the relatively short exposure periods (minutes to hours) of gamma irradiation applied in our previous studies (Table 1) can only be related to acute responses to IR, i.e., in a quasi non-metabolically active state as these studies were also performed in the dark. Therefore a number of parameters in our current study differ from our earlier transcriptomic studies on IR-exposed *Limnospira* (Table 1). First, we applied a much lower dose rate of 80 Gy·h^−1^ allowing IR exposure to extend over a full life cycle (~72 h). To attain this we had to use another irradiation facility at SCKCEN, GEUSE II. This facility operates under the same working principles as the previously used BRIGITTE and RITA facilities and consists of an irradiation container surrounded by up to 18 standard spent nuclear fuel (SNF) assemblies. Although nuclear fuels are composed of many radioactive isotopes, with a full spectrum of IR energies, the most important contribution of SNF from the BR2 nuclear reactor at SCKCEN (i.e., of one year old or older) to the gamma activity comes from ^137^Cs [14]. Second, we performed our experiment with light-emitting diodes (LEDs) as a continuous light source; hence *Limospira* cells are metabolically active in contrast to previous irradiation experiments. Third, we exposed both morphotypes mentioned above (nominated as P6 and P2 subtypes, with respectively helical and straight trichomes) of *L. indica* PCC 8005 in an attempt to associate the genomic differences between these subtypes with the different metabolic and physiological responses displayed by them when exposed to IR. Finally, fourth, we used RNA-Seq technology to overcome the intrinsic limitations of microarrays and to cover also small non-coding (nc) and regulatory RNAs.

## 2. Materials and Methods

### 2.1. Culture and Exposure

Axenic *Limnospira indica* PCC8005 cultures of helical (P6) and straight (P2) morphotypes were grown in a large volume (1 L) in an Erlenmeyer flask (Thermo Fisher Scientific, Merelbeke, Belgium) at a constant temperature of 30 °C in a Binder KBW400 growth chamber (Analis SA, Namur, Belgium), using a Heidolph Unimax 2010 rotatory shaker (Analis SA) at 121 rpm and a photon irradiance of 45 μmol photons per square meter per second (μE·m^−2^·s^−1^) produced by Osram Daylight fluorescent tubes. When cultures reached an OD_750_ of 0.5 as measured on a Genesys UV-Vis Spectrophotometer (Thermo Fisher Scientific) they were divided as triplicates into three separate volumes of 50 mL each using 250 mL Erlenmeyer flasks and subjected to a dose rate of 80 Gy·h^−1^ gamma radiation for a period of 3 days at the GEUSE II facility of SCKCEN [14].

This facility makes use of an underwater vessel surrounded by a preset number of spent nuclear fuel (SNF) rods of approximately 1-year-old. The dominant photonic energies in the applied SNF spectrum are from ^137^Cs (662 keV), with additional, minor gamma peaks originating from ^134^Cs and ^154^Eu (undisclosed BR2 reports, SCKCEN; see also [15,16]). An inbuilt LED light chamber (Figure 1) was used in the experiment for a continuous white light exposure of 45 μE·m^−2^·s^−1^ irradiance (SMD-LED warm white 1300 mcd, type NESL064AT, Nichia Corporation, Tokushima, Japan). Although the light chamber was placed on a small shaker (PSU-10i Orbital Shaker, BioSan, Riga, Letvia) to provide gentle movement of the cultures, this shaker broke down within the first 15 h of the experiment (we cannot tell at what exact cumulative dose) most likely because radiation-induced deterioration of the PIC flash memory of the display (later replacement of this module reinstalled this shaker to full operation). Yet from our experience the most determining factor for normal growth of *L. indica* in Zarrouk medium is the light source, and LEDs were unaffected by the high doses of gamma radiation. Hence, although gamma-irradiated cultures grew less well than the control cultures (grown in triplicate under irradiation-free but otherwise equal conditions), exemplified in a 15–20% lower biomass yield after 3 days, we believe this to be unrelated to the lack of agitation but mainly to be due to the prolonged exposure to gamma radiation, i.e., the increase in cumulative dose over time.

Triplicates of a non-irradiated control per dose were kept at otherwise analogous conditions in the lab. The *L. indica* P2 and P6 culture samples were collected in time intervals at three prechosen time points T1 to T3 of exposure (~15, ~40, and ~71.5 h) corresponding to approximate cumulative doses of respectively 1450, 3200, and 5700 Gy, with the actual doses for the individual samples determined by dosimetry using Harwell Amber-3042 radiation-sensitive polymethylmethacrylate dosimeters attached to the culture tubes. For sake of simplification the doses mentioned above and throughout the text for time points T1, T2, and T3 roughly correspond to the arithmetic means taken across the two series of biological triplicates; the dose rate is not constant across the exposure area inside the GEUSE II vessel, owing to the setup asymmetry and the non-uniformity of SNF rods, and due to the limitation of space inside the GEUSE II facility, P2 and P6 series of samples were irradiated at different days. Although all experimental conditions were kept as equal as possible during the two irradiation campaigns, such minor variations in the actual received doses for P2 cells versus P6 cells are inevitable. Yet we are confident that this variance does not significantly impact the outcome and interpretation of the obtained gene expression data.

### 2.2. Post-Irradiation Growth and Recovery

Small inoculants (1 mL) of irradiated and non-irradiated *L. indica* cultures were grown in 30 mL of fresh Zarrouk media in T-75 tissue flasks (Thermo Fisher Scientific). All cultures were grown in triplicates per exposed dose with their respective non-irradiated cultures under standard laboratory conditions. Recovery was followed at OD_750_ every alternate day for a period of 30 days. The proliferation curve was plotted as OD_750_ versus time using Graphpad Prism v7 (GraphPad Software, La Jolla, CA, USA—https://www.graphpad.com/)

### 2.3. RNA Extraction

The RNA extraction was performed as described before [11,12]. Three replicates of 30 mL each of the retrieved irradiated cultures and the non-irradiated control cultures were immediately put on ice after gamma irradiation and centrifuged for 20 min at 10,000 g and 4 °C, to collect the cell pellets in 15 mL conical Falcon^TM^ centrifuge tubes (BD Biosciences, Erembodegem, Belgium). Most of the Zarrouk medium was removed and resuspended cell pellets were transferred to 2 mL Eppendorf centrifuge tubes (Thermo Fisher Scientific). The remaining Zarrouk medium was entirely removed by additional centrifugation for 2 min at maximal speed. The pellets were washed three times with 1× Phosphate-Buffered Saline (PBS) and finally flash-frozen in liquid nitrogen until further analysis. A temperature of 4 °C was maintained throughout all RNA extraction procedures. Cell lysis was achieved by the RiboPure^TM^-Bacteria kit (Ambion-Life Technologies, Gent, Belgium) using Zirconia Beads in the lysis RNAwiz solution (both are kit components). The final volume of the lysis solution was adjusted according to the volume of the pellet. The released RNA was separated from cell debris by centrifugation at 10,000× *g* for 10 min at 4 °C. The further purification of the released RNA was performed with the Direct-zol^TM^ RNA Miniprep kit (Zymo Research, BaseClear Lab Products, Leiden, The Netherlands) maintaining a 1:1 ratio of organic and aqueous phase, following the manufacturer’s instructions. DNA was degraded with DNase 1 treatment (1 U/μL) and incubating at 37 °C for 30 min (Turbo DNA-free kit—Ambion-Life). Obtained RNA was concentrated with the RNA Clean and Concentrator-25 kit (Zymo Research).

The quality and integrity of the RNA were analyzed using an Agilent 2100 Bioanalyzer (Agilent Technologies, Diegem, Belgium). The RIN (RNA integrity number) value was calculated according to the manual’s instruction taking into account the ratio of two peaks of 23S rRNA (the rRNA profile of *L. indica* PCC 8005 contains three fragments instead of two, representing 16S and 23S rRNA [17]).

### 2.4. Library Preparation and RNA Sequencing

RNA sequencing was performed by NXTGNT (https://nxtgnt.ugent.be/) in collaboration with the Department of Pharmaceutical Sciences, University of Gent, Belgium). RNA quantification and quality control were performed with the Quant-iT^TM^ Ribogreen RNA Assay kit (Invitrogen) and the Agilent 2100 Bioanalyzer RNA 6000 Nano LabChip. The RiboMinus^TM^ Plant Kit for RNA-Seq (Thermo Fisher Scientific) was used for transcriptome isolation and enrichment of the whole transcriptome, through selective depletion of ribosomal RNA, according to the manufacturer’s instructions. Library preparation was done using the TruSeq Stranded Total RNA kit (Illumina, Brussels, Belgium) with fragmentation at 94 °C for 3 min instead of 8 min as to generate long fragments and with first-strand synthesis prolonged for 50 min at 42 °C instead of the normal 15 min (being adaptations to the supplier’s protocol). The libraries were amplified in an enrichment PCR with 14 cycles using standard procedures. The quality check of the libraries was performed with an Agilent 2100 Bioanalyzer High Sensitivity Chip. The libraries were quantified using a qPCR following Illumina’s Sequencing Library qPCR Quantification Guide (version of Februari 2011) and were equimolarly pooled. The pooled libraries were size-selected on a 2% E-Gel^TM^ Agarose Gel (Thermo Fisher Scientific) followed by a final library quality check on the Agilent 2100 Bioanalyzer High Sensitivity Chip. Sequencing was performed on a HiSeq 3000 system (Illumina, San Diego, CA, USA) generating 150 bp paired-end reads.

### 2.5. Data Analysis

Because the lowest cumulative dose of 1450 Gy did not seem to affect *L. indica* P2 and P6 cultures in terms of growth recovery and because the highest cumulative dose of 5700 Gy caused a much-delayed growth recovery in both strains, thus indicating massive cellular damage (as we could observe by TEM imaging in our previous study at 5000 Gy where some ultrastructures such as carboxysomes or thylakoids were disturbed or absent [13]), we decided to analyze in the first instance only RNA extracts from cultures exposed to the intermediate dose of 3200 Gy, which is at 40 h also approximately the midpoint of the organism’s lifecycle.

RNA-Seq reads obtained from these RNA extracts (both unexposed controls and exposed cultures) were aligned to the *L. indica* (formerly *Arthrospira* sp.) PCC 8005 reference genome ARTHROv5 of 2014 [9] (updated version 5 available at NCBI under GenBank assembly accession number GCA_000973065.1; also available from the MicroScope/MaGe platform [10] upon simple request to the corresponding author for conditional access) using *bowtie2* software (version 2.2.5) set at its default parameters [18]. Raw counts per gene were calculated based on the most recent genome annotation of *L. indica* PCC 8005 currently available on the MaGe platform (https://mage.genoscope.cns.fr/) [10]. Reads for coding regions were allowed to map between the start and stop codon. Where appropriate, genes in the text are described with either their gene name or using their unique identification number ARTHJROv5_XXXXX, or both.

Differential expression was calculated using the *edgeR* package (version 3.2.4) [19] in BioConductor (release 3.0, R version 3.1.2). First, the data were normalized using the weighted trimmed mean of M-values (TMM) method [20] applying the calcNormFactors()function. Next, the Cox–Reid profile-adjusted likelihood (CR) method in estimating dispersions [21] was used to take care of multiple factors by fitting generalized linear models (GLM), applying the estimateDisp() function, followed by the likelihood ratio test for differential expression analysis, applying the lmFit() and glmLRT() functions. We followed two different approaches for the definition of the contrast. The first approach consisted of four independent pairwise comparisons of the datasets P2 control (P2C), P2 irradiated (P2R), P6 control (P6C), and P6 irradiated (P6R), namely: P2 control vs. P6 control (P6C-P2C), P2 irradiated vs. P6 irradiated (P6R-P2R), P2 irradiated vs. P2 control (PR2-P2C), and P6 irradiated vs. P6 control (P6R-P6C) (in parentheses are column nominations used in Appendix A). Differentially expressed genes uniquely detected for each comparison and those in common were identified using the Venn command. For the second approach, the contrast was defined comparing irradiated samples (P2 and P6) to the control samples (P2 and P6), whilst accounting for the different strain in the design matrix (referred to as “batch normalization” in the text). The differentially expressed genes/tags were extracted by the topTags() function, and their p-values were adjusted using the False Discovery Rate (FDR) approach [22], resulting in a value for fold-change (FC) or logarithmic fold-change (log_2_FC) and a corresponding p-value corrected for multiple testing for each individual gene (Appendix A). In both approaches, genes were considered as being differentially expressed if they abided by the following selection criteria: −1 ≥ log_2_FC ≥ 1, with an FDR equal to or below 0.05.

A multidimensional scaling (MDS), commonly referred to as the Principle Coordinate Analysis (PCoA) plot, was deployed to visualize the level of similarities between the control and the radiated samples. This was done using the plotMDS() function in R with an adaption for RNA-Seq data where the distance between each pair of samples (e.g., P2C1fw and P6Rfw, etc.) is the root-mean-square deviation (Euclidean distance). For this, only the top 500 genes were retained to calculate the distance between the two samples via implementation in the Bioconductor package *limma* [23].

## 3. Results and Discussion

### 3.1. Growth-Recovery of Irradiated Cultures vs. Non-Irradiated Cultures

Both morphotypes P2 (straight trichomes) and P6 (helical trichomes) were able to return to normal growth after exposure to the three cumulative doses of 1450 Gy, 3200 Gy, and 5700 Gy. For the lowest dose at 1450 Gy, P2 and P6 cultures very closely followed the growth curves of their respective controls (Figure 2). For the intermediate-high cumulative dose at 3200 Gy, a slight delay in growth was observed in both morphotypes, with P6 taking somewhat longer to regain normal than P2. For the highest cumulative dose at 5700 Gy, P2 took six days to recover growth while P6 took up to 13 days.

This typically long lag of regaining growth after a high cumulative dose of gamma irradiation was also noted by us for the two same *L. indica* morphotypes at a higher dose rate of 600 Gy·h^−1^ [13] although with this dose rate and after a cumulative dose of 5000 Gy, P2 recovered with a long delay of 15 days while P6 did not even recover for 23 days (at which point the monitoring of regaining growth was terminated). This non-recovery of P6 at 5000 Gy (at a dose rate of 600 Gy·h^−1^) was seen by us as a diminished IR resistance of the P6 subtype owing to genomic mutations present in P6 but not in P2, or vice versa. The fact that in the current study P6 remains recoverable at the highest cumulative dose (5700 Gy) (Figure 2) may be either related to the much lower dose rate (80 Gy·h^−1^) or to other specific conditions, i.e., the presence of light (as metabolically active cells might cope better with IR) or the fact that SNF was used as a gamma source.

### 3.2. Differential Gene Expression Analysis by RNA-Seq

In order to get a first grasp of the gene networks and metabolic pathways possibly involved in the *Limnospira* response to ionizing radiation (IR) and to better understand the differences between P2 and P6 regarding IR resistance we decided to define differentially expressed genes (DEGs) by strict selection criteria: −1 ≥ log_2_FC ≥ 1 with an adjusted p-value (i.e., FDR) equal or below 0.05. This limited the number of DEGs to 1553 (887 in strain P2 and 666 in strain P6, with 352 in common) (calculated from Appendix A). For interpretation purposes, we focused on those that had, through the use of the MaGe annotation system, been given a name (e.g., *glnA*)—implying a function—or been defined as transcribing non-coding RNA. This gave a total of 229 DEGs for primary consideration (171 genes with predicted function and 58 RNA genes) (Table 2 and Appendix A).

The use of two strains P2 and P6 and two conditions, non-irradiated and irradiated, resulted in four datasets P2C, P2R, P6C, and P6R (Appendix A) which can be compared as follows: (A) differences in basal gene expression levels between P6 versus P2 before irradiation (P6C-P2C), (B) radiation-induced gene expression levels in P6 versus P2 (P6R-P2R), and (C) and (D) radiation-induced gene expression versus basal gene expression in respectively P2 (P2R-P2C) and P6 (P6R-P6C)—summarized in Appendix A for genes with predicted function and genes transcribing non-coding RNA. Such a four-way analysis may give some interesting general insights on basal gene expression across the two strains given the fact that both strains are descendants of the same ancestor and that their genomes are highly similar yet different, with 168 SNPs, 48 indels, and four large insertions affecting a total of 41 coding regions across both genomes [13]. Yet, it remains difficult to compare gene expression profiles between P2 and P6 as gene expression in either strain may be directly or indirectly affected by said genomic differences. In fact, the gene expression patterns for non-irradiated P2 and P6 are not equal, with 225 genes across the two strains showing different levels of expression as scored by the same stringent selection criteria as for “induced” or “repressed” genes in the same organism, i.e., −1 ≥ log_2_FC ≥ 1 and FDR ≤ 0.05 (calculated by Microsoft Excel COUNTIF operations in Appendix A). To normalize these slightly variant expression patterns between P2 and P6, housekeeping genes could be used to apply a multifactorial statistical correction (i.e., using the expression levels of a set of reference genes). For cyanobacteria, a number of genes have been recently suggested as reference genes in qPCR transcriptomic studies [24,25,26]. The log_2_FC[P6C-PC2] values for these genes (Appendix A, summarized in Appendix A) generally confirm that the difference between the expression patterns for P2 and P6 remains sufficiently low, with a FC value for most of these reference genes around 1 albeit with FDR values > 0.05. The outliers in this set are the two *rrnB* genes encoding 16S rRNA (*L. indica* PCC 8005 has two copies of the 16-23S rRNA operon), both with FC values of 0.61, and also *secA*, with an FC of 0.69. However, rRNA levels were lowered significantly in the RNA purification procedures via rRNA depletion (see methods Section 2.4) rendering differential expression data for the *rrn* genes in Appendix A meaningless. Additionally, the use of the *rrnB* gene as a reference gene in bacterial transcriptomics is controversial since rRNA and mRNA are degraded at different rates [27]. Furthermore, the copy number of *rrnB* can be much higher than for other genes [28]. Unsurprisingly the above three studies [24,25,26] showed that, for a number of cyanobacteria and for a variety of conditions, the *rrnB* gene may not be a good choice for the normalization of transcriptomic data. In addition, these studies also showed that *secA* did not perform well as a reference gene, at least for some cyanobacteria under some conditions.

For these reasons we considered the genetic background of each strain as a “batch” condition and performed batch normalization (see Methods), resulting in a set of 660 DEGs (Appendix A) with 98 DEGs having a predicted function (Appendix A and Table 3, Table 4 and Table 5) and 32 DEGs transcribed into non-coding RNA (Appendix A and Table 6, Table 7 and Table 8). All 130 (98 + 32) genes but one verified in this way belong to a subset of the 229 (171 + 58) genes selected prior to batch normalization. The only exception being *narH* (ARTHROv5_10325) (in fact a mere gene fragment and identified by MaGe as being an fCDS) which was not seen as a true DEG in the original P2/P6 comparison (Appendix A: [P2R-P2C] → log_2_FC 1.73, FDR 0.087; [P6R-P6C] → log_2_FC 0.66, FDR 0.47), yet was scored as a DEG after batch normalization (Appendix A: log_2_FC 1.12, FDR 0.048, bringing the total in this table inadvertently to 99). We discuss the majority of the 130 DEGs verified via batch normalization in separate sections below. Note that for all these genes the original FC is displayed rather than the FC after batch normalization as to allow comparison between P2 and P6 expression profiles. FC values and trends across the two approaches, i.e., prior and after normalization, are highly similar and fully corroborate to each other (verifiable with Appendix A).

#### 3.2.1. Genes Regulated by γ-Radiation in Strain P2 But Not in Strain P6

In the P2 morphotype (straight trichomes) of *L. indica* PCC 8005, a total of 887 genes were differentially expressed by exposure to gamma radiation (336 upregulated and 551 downregulated) (Table 2). Out of those, 119 had a defined function according to the MaGe annotation platform (42 upregulated and 77 downregulated). Additionally, 43 genes were transcribed into non-coding RNA (21 upregulated and 22 downregulated) (Table 2). Verification with batch normalization resulted in 19 genes (8 induced, 11 repressed) only regulated in P2 but not in P6 (indicated in Appendix A and listed separately in Table 3).

The *mutT1* gene (_40086) encodes a 8-oxo-dGTP diphosphatase/NUDIX hydrolase that helps to rid the cell of ROS-oxidized nucleotides which are highly mutagenic as they cause errors in DNA replication. The genome of the model organism for radiation resistance *D. radiodurans*, contains at least 23 genes encoding such 8-oxo diphosphatase/hydrolases, some of which may act to “sanitize” other mutagenic (radiation-evoked) DNA precursors [29]. In *L. indica* PCC 8005, four other *mutT* genes exist (_30367, _30835, _60942, and _61161) but these were not scored as DEGs either in P2 or P6.

The SigG sigma factor encoded by the *sigG* gene is ubiquitous to all cyanobacteria and belongs to the so-called extracytoplasmic function (ECF) family of alternative sigma factors. Members of this family receive specific external stimuli to control the expression of proteins residing in the outer membrane or periplasmic space and hence are able to swiftly react to adverse conditions including high-intensity light, UV radiation, salinity, desiccation, antibiotics, and heavy metals. Although the strict DEG selection scores sigG only induced in P2, it is worthwhile to note that this gene has an FC of 1.93 (FDR = 0.012) in strain P6 (Appendix A).


microorganisms-09-01626-t003_Table 3Table 3Irradiation-induced and repressed genes of known function in P2 but not in P6.GeneMaGe-IDPredicted FunctionCOG-IDClassFC
***czcD***
10962cation efflux system proteinCOG1230P2.71
***mutT1***
40086NUDIX hydrolase, MutT-like mutator proteinCOG1051F2.38
***sigG***
40126RNA polymerase sigma factor, ECF subfamilyCOG1595K2.92
***acaE***
40592precursor peptide (cyanobactin), PatE-likendnd2.73
***nanE***
41334N-acylglucosamine-6-phosphate 2-epimeraseCOG3010G2.42
***sseA***
600263-mercaptopyruvate sulfurtransferaseCOG2897P2.53
***pflB***
60899pyruvate formate lyase ICOG1882C2.78
***isiA***
61180iron stress-induced chlorophyll-binding proteinndnd2.32
***hisR***
30044transcriptional 2-C system response regulatorCOG0745T0.35
***insB***
30106transposase InsAB′, IS*1* family (fragment)COG1662L0.30
***rfpX***
30213fluorescence recovery protein (RFP)ndnd0.18
***faxB3***
30751tentative phage proteinndnd0.14
***hliA***
40644high light-inducible protein (HLIP)ndnd0.18
***chlN***
41145protochlorophyllide reductase subunitCOG2710C0.31
***faxB4***
50359tentative phage proteinndnd0.17
***kaiA***
60140circadian clock proteinndnd0.39
***kaiB***
60141circadian clock proteinCOG0526C, O0.27
***dam***
60398DNA adenine methylaseCOG0338L0.41
***corA***
60812magnesium/nickel/cobalt transporterCOG0598P0.43MaGe-ID, unique gene identifier of the MaGe Genomes Database (https://mage.genoscope.cns.fr/) [10] for ARTHROv5; COG-ID, Database of Clusters of Orthologous Genes (COG) Definition [30]; class, classification of COGs into functional categories (one-letter codes explained in Appendix A); FC, fold change; Induced (green): FC ≥ 2, Repressed (red): FC ≤0.5; all genes abide to the selection criteria of |log_2_FC| ≥ 1 and FDR ≤ 0.05 (see Methods), highly induced or repressed genes (four-fold or higher) are indicated with deeper green or red, respectively.


The *sseA* gene encodes a 3-mercaptopyruvate sulfurtransferase (3-MST) that may be involved in cysteine and methionine metabolism, tRNA sulfuration, and the generation of sulfane sulfur species that may help to protect cells against oxidative stress. MSTs are ubiquitous across all domains of life yet only very few prokaryotic MSTs have been structurally and biochemically characterized [31,32] while their function in cyanobacteria remains enigmatic.

The *isiA* gene codes for a CP43-like chlorophyll-binding protein that acts as an antenna protein under iron-limiting conditions, protects the PSI photosystem at high-light irradiances by forming a large protective multi-subunit ring-shaped complex around PSI, and has a great capacity to dissipate excesses of excited-state energy, hence preventing over-excitation of PSII (reviewed in 2018 by Chen and colleagues [33]). Recently, it has been proposed that the actual major function of the IsiA pigment–protein complex would be to act as a storage depot for up to 50% of the cellular chlorophyll content during stress-induced degradation of phycobilisomes which effectively prevents cells to absorb light under conditions of metabolic arrest [34]. In this context, the IR-induced expression of the *isiA* gene makes sense: not only does it serve to dissipate excesses of energy, but it also keeps a chlorophyll pool ready for use in the post-irradiation recovery phase. The fact that *isiA* is induced by gamma-irradiation in P2 but not in P6 (or at least not as distinctively, with an FC = 1.81 and an FDR = 0.074 hence not being scored in P6 as a DGE) (Appendix A) may explain in part the somewhat faster recovery of P2 cells after irradiation-free regrowth in fresh medium (Figure 2).

Among the genes repressed uniquely in strain P2, *rfpX* and *hliA* are of immediate interest (Table 3). The former encodes a Fluorescence Recovery Protein (FRP), a small protein of 106 aa that exists in dimeric and tetrameric forms and in natural conditions plays a crucial role in cyanobacteria for the protection against the adverse effects of high-intensity light (HL) [35,36]. This protection is essential because longer periods of intense light inevitably will lead to a saturation in the cell’s capacity for photosynthesis and in turn, will increase the levels of reactive oxygen species which damage pigments, lipids, and PSI and PSII proteins of the photosynthetic thylakoid membrane [37] (and references therein). The latter encodes an HL-inducible protein. Such proteins are mostly located in the PSII system and have not only a chlorophyll-protein protective function but also an energy-quenching role [38]. It is odd that these two genes, *rfpX* and *hliA*, are firmly repressed (five-fold) by irradiation in the P2 strain which is known to grow slightly better under standard conditions and also recovers better from gamma irradiation. One would think that gamma rays, which have extremely high photonic energies, would elicit the opposite effect and cause a higher—not lower—expression of these two genes. Importantly, neither *rfpX* nor *hliA* was identified as gamma radiation-regulated in previous studies [11,12], which in fact confirms our results for the P6 strain. Hence, the tight repression of these genes in the irradiated P2 strain deserves detailed follow-up experiments with gene-specific RT-qPCR analyses.

Interestingly, also the *kaiABC* circadian locus was well repressed in P2 but not regulated in P6 [note that although *kaiC* is not seen as a DEG in the normalization procedure (Appendix A) it was registered as a DEG in the original comparison, being repressed more than two-fold in P2 but unregulated in P6 (Appendix A and Table 3)]. The KaiABC circadian clock—essentially measuring time in 24 h periods—enables an organism to regularly coordinate and adjust its cellular processes including major steps in its cell cycle and key metabolic functions [39,40]. In cyanobacteria, a number of additional genes are involved in circadian expression, i.e., *rpaA*, *rpaB*, *sasA*, *labA*, *cdpA*, *cpmA*, *ldpA*, *ircA*, *prkE*, *lalA*, and *cikA* [41,42]. This spurred us to look for these genes in the *L. indica* PCC 8005 genome using the *Synechocystis* sp. PCC 6803 counterpart protein sequences as queries for BLAST searches against the *L. indica* PCC 8005 proteome at MaGe (ARTHROv5) [10]. All these genes could be found in the PCC 8005 genome, with their gene products displaying between 30 and 87% sequence identity with their query. Only *pex* (_20131) and *sasA* (_60943) were correctly named in the MaGe annotation platform and hence were considered in our analyses as genes with predicted function, while none of the other genes (*rpaA*, _12022; *rpaB*, _60282; *cdpA*, _41365 and _41035; *cpmA*, _20263; *ldpA*, _11956; *ircA*, _40296; *prkE*, _41401 and 40698; *lalA*, _40200) were named as such in MaGe and thus did not show up in our analyses beyond Appendix A. When we checked the full list of genes (including *pex* and *sasA*) using their unique protein identifier for regulation by gamma-irradiation (Appendix A) only the *cikA* gene (_41335) showed up. This gene is, like the *kaiABC* locus itself, more than two-fold repressed in P2 (log_2_FC = −1.31, FDR = 0.002) and not regulated in P6. The CikA protein is a histidine kinase with roles in time entrainment (i.e., a clock reset in the cue of environmental changes), output signalling, and cell division [40,43]. Several studies on a variety of cyanobacteria have shown that the circadian system (with the core clock constituted by the KaiABC complex and the three input/output proteins SasA, CikA, and RpaA) controls gene expression at a global cell scale regulating a large portion of their genome in the range of 20 to 79% [41,44,45]. In addition, in cyanobacteria the circadian clock needs to work unperturbed as to ensure complete chromosome replication [46]. Thus, although the reasons why *kaiABC* and *cikA* gene expression is repressed by gamma irradiation in *L. indica* P2 but not in P6 remain elusive for now, it is clear that any disturbance in P2 circadian rhythm will bear a cell-wide impact on many cell processes, possibly explaining or augmenting the different routes taken by P2 and P6 in coping with IR.

The *dam* gene encoding the *L. indica* DNA adenine methylase is also more than two-fold repressed in P2 but not regulated in P6. This gene (_60398) is not associated with any of the restriction–modification (RM) systems in the *L. indica* genome. Such “orphan” MTases are widespread among bacterial genomes [47] and it has been recognized that Dam methylation plays an important role in the regulation of bacterial gene expression and DNA repair and replication [48,49]. It is possible that differences in *dam* gene regulation between strains P2 and P6 give rise to different Dam methylation patterns in their genomes which in turn may help explain in part the variance in the IR response routes deployed by these strains.

#### 3.2.2. Genes Regulated by γ-Radiation in Strain P6 But Not in Strain P2

In the P6 morphotype (helical trichomes) of *L. indica* PCC 8005, a total of 666 genes were differentially expressed by exposure to gamma radiation (398 upregulated and 268 downregulated) (Table 2). Out of those, 114 had a defined function according to the MaGe annotation platform (55 upregulated and 59 downregulated). Additionally, 31 genes were transcribed into non-coding RNA (9 upregulated and 22 downregulated) (Table 2). Verification with batch normalization resulted in 14 genes (9 induced, 5 repressed) only regulated in P6 but not in P2 (indicated in Appendix A and listed separately in Table 4).


microorganisms-09-01626-t004_Table 4Table 4Irradiation-induced and repressed genes of known function in P6 but not in P2.GeneMaGe-IDPredicted FunctionCOG-IDClassFC
***cry***
10963deoxyribo-dipyrimidine photolyaseCOG0415L2.66
***groL2***
30259chaperonin GroEL, large subunit LCOG0459O11.13
***psbI***
30303photosystem II reaction center proteinndnd2.41
***cbsR***
30501transcriptional regulator (cysteine biosynthesis)COG0664T3.13
***cysA***
30503sulfate/thiosulfate import ATP-binding proteinCOG1118P2.97
***cas2***
40676CRISPR-associated endoribonucleaseCOG1518L2.42
***proA1***
41057γ-glutamyl phosphate reductaseCOG0014E2.80
***cyp***
60259cytochrome P450COG2124Q2.99
***cheY1***
60578response regulator (receiver domain), 2-C systemCOG0784T3.49
***glnA***
12133glutamine synthetaseCOG0174E0.24
***ntcB***
30796transcriptional activator (nitrogen assimilation)COG0583K0.46
***hypB1***
40489hydrolase (nickel liganding into hydrogenases)COG0378K0.33
***nblB1***
50028phycocyanin α-phycocyanobilin lyaseCOG1413C0.34
***nthA***
60175nitrile hydratase α subunitndnd0.30(abbreviations, colors, and selection criteria are as in Table 3).


Immediately standing out in the list of P6-specific DEGs is the chaperonin-encoding *groL2* gene (_30259) which is induced over ten-fold in response to γ-radiation (FC = 11.1). While this gene is solitary placed on the genome another copy of the gene, *groL1* (_61181), is accompanied by its cochaperonin-encoding *groS* gene (_61182). Chaperonins promote protein folding and are known to play a role in the maintenance of cellular stability under a wide variety of stress [50]. Though most cyanobacteria encode one *groSL* locus and one additional monocistronic *groL* many also contain a second *groSL* [51]. The *L. indica* PCC 8005 proteins GroL1 and GroL2 are of nearly the same size (545 and 558 aa, respectively) and are 64% identical on peptide level. As chaperonins normally require an interaction of the large (L) and small (S) subunits to function properly, it is possible that GroL1 and GroL2 compete for the same GroS partner. Alternatively, GroL2 may have evolved a specialized function while GroL1 kept a housekeeping function [52]. Note that the *groSL1* locus (_61181/2) is induced in both P2 and P6 (Table 5) but where *groSL1* expression is only 2–3 fold elevated in P2, it is massively induced, ca. 30-fold, in P6. It is tempting to speculate that P6 proteins are more heavily damaged by gamma irradiation than P2 cells—which would be in line with the noted difference in IR resistance between the two strains—and therefore require more abundant levels of GroSL chaperonins, whether of mono- or bicistronic origin. Reversely, the P2 strain may have either lost the ability to induce these heat shock genes or simply does not need the strong induction of these genes as it incurred lesser damage than P6. Yet the P2-P6 orthologous coding and/or regulatory sequences for those genes are deemed identical based on whole-genome sequencing [13], so the remarkable variance in *groSL/L* gene induction between P2 and P6 with roughly one order of magnitude must be attributed to genetic pleiotropy involving unknown proteins, signal molecules, or ncRNAs. A preliminary analysis of the −200 upstream regions of the *L. indica* PCC 8005 bicistronic *groSL1* and monocistronic *groL2* loci learns that both regions contain a consensus CIRCE element (Controlling Inverted Repeat of Chaperone Expression) which has been shown in a variety of bacteria to act as a negative cis-element bound by HrcA (Heat shock regulation at CIRCE). However, the *hrcA* gene (_40278) in our RNA-Seq analysis was not regulated, so other regulatory mechanisms for gamma radiation-related induction of *groSL/L* might be involved. A number of additional regulatory sequences have been discovered in duplicate *groSL/groL* upstream regions across many prokaryotes, elucidating a distinct regulation of these gene loci including novel modes of light-responsive regulation [53,54]. So far we detected a light-responsive K-box element in the *groSL1* promotor region but not in the *groL2* promotor region. Clearly, a more detailed analysis on these groSL/L loci is called for, including time course studies by locus-specific qRT-PCR on *L. indica* P2 and P6 cells subjected to γ-radiation.

The induction of the *cry* gene (_10963) in P6 but not in P2 cells is of interest as this gene encodes a deoxyribo-dipyrimidinephotolyase cryptochrome (Lin-CRY) with the ability to repair cyclobutane pyrimidine dimer (CPD) lesion for both single-strand (ss) DNA and double-strand (ds) DNA [55]. Such CPD lesions are typically incited by UV as part of the solar light spectrum and photolyases are photon-triggered enzymes that revert this type of damage without relying on *de novo* DNA synthesis [56]. In our experiments, we only used LED lighting with an emission spectrum above 400 nm (see Methods) and hence the 266% induction of *cry* gene expression in the P6 strain cannot be UV-related. Additionally, gamma photons are far more energetic than UV photons and generally cause a different type of damage either directly resulting in ss and ds strand breaks or indirectly via the generation of ROS causing oxidative DNA damage, in both cases calling for other DNA repair systems. Still, it is possible that Lin-CRY with its unique ability to repair dsDNA CPD lesions and a unique methenyltetrahydrofolate (MTHF) chromophore-binding pattern, has yet unidentified activities related to γ-radiation-induced DNA damage and cellular responses, warranting further investigations. Interestingly, the *Synchocystis* PCC 6803 homolog Syn-CRY, in a sequence 62% identical to Lin-CRY, has been shown to have a specific physiological role in PSII repair [57]. In this context it is worth mentioning that the 38 aa gene product of *psbI*, seen as a DEG in P6 but not in P2 (Table 4), is thought to be involved in PSII assembly and also repair through interaction with the D1 and CP43 proteins [58,59], D1 being essential for PSII function—and constantly in need of replacement because it is particularly susceptible to photoinduced damage—and CP43 being a core light-harvesting pigment–protein complex.

In strain P6, the cytochrome P450 gene *cyp* is strongly induced (FC = 3; Table 4). Cyanobacterial CYP monooxygenases play a crucial diversifying role in the production of secondary metabolites because of their regio- and stero-specific oxidation of a range of substrates [60]. Since some of these metabolites may have antioxidant or photo-protective properties, the induction of CYP in response to IR could make sense. Yet, such a CYP induction may imply a considerable investment in metabolic terms, something the already IR-stressed cells may not be readily able to afford. The more cautious CYP response in strain P2 (an FC of 1.8 and FDR of 0.033) may thus be a more favorable trade-off, in line with its better growth recovery from IR exposure.

The *cysA* gene displaying a 3-fold induction by SNF γ-irradiation in the P6 strain (Table 4) encodes a sulfate-transporting ATPase and is part of a gene cluster *cysARPWT* (_30503 to _30507), with CysR a transcriptional regulator and CysPWT constituting an ABC transporter system. In our study, neither *cysR* nor *cysPWT* was regulated in P2 or P6 (although *cysP* was scored as a DEG prior to normalization with an FC of 2.51 and an FDR of 0.007—Appendix A). Because we worked with strict DEG selection criteria, *cysA* was not listed as a DEG in P2 because of an FDR of 0.052 yet it displayed a solid 2-fold induction (Appendix A). It is possible that under radiation stress, *L. indica* attempts to enhance sulfate uptake as it is in dire need of sulfur in glutathione biosynthesis (with cysteine as a precursor), in thiol groups of antioxidant enzymes (e.g., thioredoxins), in other thiol-disulfide exchanging proteins and ROS-signalling enzymes containing a Cys-X-X-Cys active site, or in the many key sulfur-containing compounds in the cell (i.e., sulfolipids, vitamins like biotin and thiamine, co-factors, etc.). Such cellular need for adequate levels of sulfur is also in line, at least in P2, with the increased production of 3-mercaptopyruvate sulfurtransferase involved in the cellular production of L-cysteine and encoded by *sseA* (previous section, Table 3). Immediately downstream of *cysA* lays another gene, *cbsR* (_30501), encoding a CRP/FNR family type regulator. This *cbsR* gene is induced in P6 over 3-fold (Table 4) and is followed by four genes *cysK2 cysE1*, *srpI*, and *sufS2* (_30500 to _30497) encoding a cysteine synthase, a serine O-acetyltransferase, a major membrane protein, and a cysteine desulferase, respectively, with *cysK2* one of three cysteine synthase genes, *cysE1* one of two serine acetyltransferase genes, and *sufS2* one of two cysteine desulferase genes present in the *L. indica* PCC 8005 genome, exemplifying the importance of its sulfur biogenesis and cysteine production. The observed repression of *cysP* and *sseA* only in P2, the upregulation of *cysA* in P6 (and likely P2) and the upregulation of *cbsR*, only in P6, are clear signs that the P2 and P6 strains have to cope, in response to IR exposure, with specific limitations and capacities in their sulfur households (see also our discussion in Section 3.2.3 on the commonly regulated *metE* gene).

As mentioned above, *cysE1* encodes a serine O-acetyltransferase, an enzyme catalyzing the formation of O-acetyl-L-serine (OAS) from L-serine. This OAS forms the amino acid skeleton for the production of cysteine with the input of free sulfides, interconnecting sulfate, nitrogen, and carbon assimilation in the cell. Looking at Table 4 for repressed genes in P6 but not in P2 one immediately notices the tight repression of the *glnA* gene, with an FC equal to 4.2. This gene encodes glutamine synthetase, an essential enzyme in nitrogen metabolism that catalyzes the condensation of glutamate, a pivotal carbon skeleton, and free ammonia to form glutamine. This confirms our previous findings [11,12] when we reported an immediate and full shutdown of *glnA* expression in *L. indica* PCC 8005 cells exposed to high doses of ^60^Co-gamma radiation. Glutamine synthetase (GS) in cyanobacteria features regulatory systems that are very different from those of most prokaryotes (reviewed in 2018 by Bolay and colleagues [61]): (i) cyanobacterial GS interacts with one of two small inhibitory peptides of 7 and 17 kDa, the so-called inactivating factors (IFs) IF7 and IF17, that fully block GS activity at their highest concentrations, (ii) *glnA* and the genes encoding IF7 and IF17 (*gifA* and *gifB*, respectively) are, amongst other genes, controlled by NtcA, a global transcriptional regulator in nitrogen- and carbon metabolism that can act as a repressor or activator depending on the location of its binding site, and iii) IF abundance is tightly tuned by small non-coding (nc) RNAs that interfere with gene-specific transcript translation, some of which need to bind to glutamine (to so-called glutamine riboswitches that are unique to cyanobacteria) to obtain their most interfering secondary structure. In the MaGe database for *L. indica* PCC 8005, no *gifA* or *gifB* genes were annotated as such (and hence not taken into account in our original analyses), requiring BLASTp searches against the PCC 8005 proteome with the *Synechocystis* sp. PCC 6803 GifA and GifB sequences (Ssl1911 and Sll1515, respectively). This search yielded four potential *gifA* genes (_60802 to _60805) and two potential *gifB* genes (_11960 and _41129). The _11960 gene (now called by us *gifB1*) is immediately preceded by glutamine riboswitch RNA94. This resembles the situation in Synechocystis sp. PCC 6803 where the *gifB* (*sll1515*) gene is transcribed together with a 104 nt long untranslated transcribed region (5′UTR), containing the predicted *glnA* aptamer [62]. The other gene _41129 (provisionally called by us *gifB2*) does not have such a sequence in its 5′UTR. A second glutamine riboswitch was found in the PCC 8005 genome as gene RNA199. None of these genes were scored as DEGs in our analyses prior to normalization (Appendix A) (and not withheld after normalization—not all data shown). Nonetheless, we should note that in our original analysis *gifA2*, *gifA4*, *gifB1* and both riboswitches were 165–195% up- or downregulated in strain P2, each with an FDR value below 0.05 (except RNA199 with an FDR of 0.062), yet were unregulated in strain P6 (Appendix A).

Although the global nitrogen regulator NtcA (which in *Synechocystis* sp. PC 6803 activates genes such as *glnA*, *glnB*, *nirA*, and *narB*, amongst others, and represses *gifA* and *gifB* [63]) was previously shown by us to be repressed by high doses of ^60^Co-gamma radiation [11], it was not regulated in our current analysis. The *glnB* gene encoding the PII signal transducer protein playing a central role in the modulation of carbon- and nitrogen metabolism-related processes and the regulation of ammonium, nitrate/nitrite, and cyanate uptake [64], is repressed in P6 but not in P2 as observed prior to normalization (Appendix A; FC = 2.5/FDR = 0.000) and marginally not seen as such after normalization (FC = 1.95/FDR = 0.000; Appendix A). In *Synechocystis* sp. PCC 6803, PII controls ammonium uptake by interacting with the Amt1 ammonium permease and mediates nitrate uptake by interacting with the NrtC and NrtD subunits of the nitrate/nitrite ABC-transporter NrtABCD [64]. In our study, prior to normalization, *amt1* was like *glnB* scored as a repressed gene in P6 but not in P2 (Appendix A; FC = 2.38/FDR = 0.000) yet it was not retained as such after normalization (FC = 1.55/FDR = 0.03; Appendix A). Nonetheless, the *nrtABCD* locus is firmly repressed in both P2 and in P6 before and after normalization (Appendix A and Table 5). Additionally, the *nrtP* gene encoding an MSF family nitrate transporter and the adjacent *narB* gene encoding a nitrate reductase, as well as the ferredoxin-nitrite reductase gene *nirA,* are tightly repressed in both strains P2 and P6 (Table 5). Likewise, the *cynBDX* genes encoding a putative cyanate transporter (or at least parts thereof) and the cyanase encoding *cynS* are highly repressed in both strains P2 and P6 (Table 5—*cynX* was manually added afterwards as it was previously unnamed but is clearly part of the *cynBDXS* gene cassette and was validated as a DEG after normalization, with FC = 3.5 and an FDR = 0.000). Two unnamed gene fragments (_11875/6) upstream of *cynB* appeared to be part of this cassette as they form one single gene in other sequenced *Arthrospira/Limnospira* genomes (MaGe database [10]) as well as in other cyanobacterial genomes [65]. Together they encode a substrate-binding protein similar to NrtA/CynA. Although additional analysis is required to establish whether these gene fragments are the result of a mutation or sequencing error in the PCC 8005 genome, both genes were firmly repressed in both P2 and P6 before (Appendix A) and validated as DEGs after normalization (Appendix A). Two more nitrogen-related genes scored as a DEG and repressed in P6 but not in P2 are the *ntcB* (_30796) and *nthA* (_60175) genes (Table 4). The former encodes a LysR-type, nitrite-responsive transcriptional regulator which is specifically involved in the activation of genes involved in nitrate assimilation (e.g., *nirA*, *narB*, *nrtABCD*, *nrtP*, etc.) [66]. The latter encodes the nitrile hydratase alpha subunt and is accompanied by *nthB* (_60176) for the beta unit as well as the *nthE* (_60174) gene encoding an NthAB activator protein. Nitrilate hydratases are able to free nitrogen from organic nitriles (R–C≡N) and thus open up, next to the ammonium/nitrate/nitrite and cyanate routes, an additional route for nitrogen assimilation. The *nthB* gene was firmly repressed in both P2 and P6 (Appendix A and Table 5) while *nthE*, like *nthA*, was only repressed in P6 (Appendix A—note that gene _60174 is only named afterwards as *nthE* and thus was not present in our analyses). Taken together, downregulation of nitrogen assimilation ran quite similar in the P2 and P6 morphotypes of *L. indica* PCC 8005 and was very much in line with our previous studies [11,12], with most of the involved genes repressed in both. Nevertheless, *glnA* (and probably also *glnB*), *ntcB*, *nthA* and *nthE* were clearly regulated in a strain-specific way, with a potential impact on cellular pathways and IR-incited responses.

#### 3.2.3. Genes of Strains P2 and P6 Commonly Regulated by γ-Radiation

Of the 1553 genes regulated by γ-radiation across P2 and P6, 352 genes were regulated in both strains (Table 2). Of those, 62 had a defined function according to the MaGe annotation platform (28 up- and 34 downregulated, with four genes added afterwards to Table 5—see text).

The *rnc2* gene (_10310) encoding a ribonuclease III is highly induced by γ-radiation in both P2 and P6 (Table 5). Such RNases are involved in RNA processing and microRNA generation [67]. Recently, RNase III was also implicated in global gene expression in the cyanobacterium *Synechococcus* sp. strain PCC 7002 [68] whose genome harbours three RNase III homologs (A0061, A2542, A0384). A second *L. indica* RNase III-encoding gene, *rnc1* (_30253) was repressed in P2 (FC > 2, FDR = 0.000) but not regulated in P6 cells in our original analysis prior to normalization (Appendix A), after which it was not withheld as a DEG (Appendix A). The *rnc1* and *rnc2* products were 49% and 57% identical to A2542 and A0061, respectively, but a third homolog corresponding to the *Synechococcus* A0384 “Mini-RNase III” was not found in the *L. indica* PCC 8005 proteome (via BLASTp using A0384 as a query). It has been suggested that the *Synechococcus* A0061 and A2542 RNase III play a role in processing pre-23S-rRNA explaining the significant alterations in the genome-wide expression patterns of single and combined ΔA2542/Δ0061 mutants [68]. Seen in this context, the high induction of *rnc2* in response to γ-radiation in both P2 and P6 might be related to switches and rerouting of global protein expression and hence increased needs in RNA degradation, maturation and processing.

An interesting pair of genes commonly induced in both strains P2 and P6 are *norB* and *glbN* (_10323/4). The former gene encodes nitric oxide reductase (NOR) that should be regarded as a detoxifying enzyme as it converts the reactive nitrogen species (RNS) nitric oxide (NO) to the lesser reactive nitrous oxide (N_2_O) while *glbN* encodes a cyanoglobin able to bind, as all hemoglobins do, oxygen with high affinity but in a reversible manner [69]. In bacteria, NO levels must be carefully monitored and regulated because it is involved in many signaling networks and physiological conditions. In addition, NO is a reactive molecule that has the ability to attack, like ROS and other RNS, cellular components and requires active management. Cyanoglobins not only have a high affinity to oxygen (they probably act as oxygen scavengers) but also bind NO and as such may be key participants in the nitrogen–oxygen chemistry of cyanobacterial cells. What intrigues is the apparent genetic linkage between *norB* and *glbN* in the *L. indica* PCC 8005 genome and future investigations should include sequence analysis of *norB* and *glbN* upstream regions (URs) to identify regulatory sequences. For instance, *glbN* transcription is controlled by NtcA in Nostoc sp. UTEX 584 [70], and additional *norB-glbN* IR-induction experiments would help us to fully appreciate the functional role of a GlbN cyanoglobin in *Limnospira*‘s resistance to ionizing radiation.

Of the four intact and probably active *dnaK* genes in the *L. indica* PCC 8005 genome, i.e., *dnaK1* (_30014), *dnaK2* (_30686), *dnaK4* (_11814), and *dnaK5* (_10362), of respectively 530, 697, 658, and 737 aa in size, only the *dnaK5* gene is scored as a DEG in our analyses and was found to be highly induced (i.e., four- to sixfold) by γ-radiation in both strains P2 and P6 (Table 5). The DnaK protein is the bacterial equivalent of the eukaryotic heatshock protein Hsp70 and plays a crucial role in protein stability and folding under a variety of stress conditions and handles protein-targeting and protecting functions in non-stressed cells [71]. It is estimated that in *E. coli* up to 25% of all cytoplasmic proteins interact with DnaK [72]. The occurrence of multiple *dnaK* genes in cyanobacterial genomes is rather common and indications are that they exist and function in various cellular compartments and have specific expression profiles [73,74]. The considerable induction of the *L. indica* PCC 8005 *dnaK5* gene in both P2 and P6 upon exposure to gamma radiation certainly warrants further investigation.

A striking set of genes commonly induced in both strains P2 and P6, are the five *arh* genes A to E (_10467 to _10471) (Table 5). These genes were strongly (i.e., 8 to 30-fold) induced by SNF-gamma irradiation, confirming our previous reports on ^60^Co-gamma irradiation of *L. indica* with induction levels of the same order [11,12]—please note that these genes since then were renamed so that *arhA* became the first gene and *arhE* the last. An updated BLASTp search against the GenBank Non-Redundant Protein Sequence Database (NRDB) of May 2021 did not result in any new information in regard to their function. All we know so far is that these five genes are most likely co-transcribed (on the basis of short, ostensibly promotor-less intergenic regions) and are very likely under control of an XRE-type transcriptional regulator encoded by the *arhR* gene (_10466) immediately preceding *arhA* and transcribed into the opposite direction.


microorganisms-09-01626-t005_Table 5Table 5Irradiation-induced and repressed genes of known function common to P2 and P6.GeneMaGe-IDPredicted FunctionCOG-IDClassFC (P2 and P6)
***rnc2***
10310ribonuclease III (16S/23S rRNA formation)COG0571K4.716.58
***norB***
10323nitric oxide reductase subunit BCOG3256P2.722.02
***glbN***
10324cyanoglobin (hemoglobin)COG2346R4.712.46
***narGb***
10336nitrate reductase, alpha subunit (fragment)COG5013C11.5610.48
***dnaK5***
10362chaperone protein (Hsp70 equivalent)COG0443O6.054.28
***arhA***
10467conserved hypothetical proteinndS12.718.37
***arhB***
10468conserved hypothetical proteinndS29.8511.33
***arhC***
10469conserved hypothetical proteinndS23.7915.10
***arhD***
10470conserved hypothetical proteinndS23.3615.65
***arhE***
10471conserved hypothetical proteinndS22.2018.12
***phaP***
10501phasin (54% aa identity with ssl2501)ndnd3.082.63
***ubiA1***
108544-hydroxybenzoate octaprenyltransferaseCOG0382H2.622.50
***rmlA***
12054glucose-1-phosphate thymidylyltransferaseCOG1209M2.052.14
***dusA***
20088tRNA-dihydrouridine synthase ACOG0042J4.903.75
***panE***
305912-dehydropantoate 2-reductaseCOG1893H3.842.93
***hsdR1a***
30623Type I site-specific deoxyribonuclease (part)COG0610V8.0014.03
***hsdR1b***
30624Type I site-specific deoxyribonuclease (part)COG0610V4.495.09
***hsdR1c***
30625Type I site-specific deoxyribonuclease (part)COG0610V2.192.31
***cas1***
40678CRISPR-associated endonuclease Cas1COG1518L2.572.89
***folE1***
40925GTP cyclohydrolase ICOG0302H5.514.83
***pyrD***
41290dihydroorotate dehydrogenaseCOG0159E2.252.07
***cheC1***
60571inhibitor of MCP methylationCOG1776N2.022.42
***cheB1***
60572chemotaxis protein methyl-esteraseCOG2201N3.213.08
***cheW1***
60576purine-binding chemotaxis proteinCOG0835N2.174.47
***cheA1***
60577signal transduction histidine kinaseCOG0643N2.593.66
***metE***
60603homocysteine methyltransferaseCOG0620E7.624.94
***ppiC***
60867peptidylprolyl isomeraseCOG0760O2.372.52
***groL1***
61181chaperonin GroEL, large subunit LCOG0459O2.8229.87
***groS***
61182chaperonin GroEL, small subunit SCOG0234O2.3732.34
***stpA***
10080glucosylglycerol 3-phosphatasendnd0.320.44
***yhdJ***
10381DNA adenine methyltransferaseCOG0863L0.460.39
***livG***
10485leucine/isoleucine/valine transporter componentCOG4674R0.310.31
***nadC***
10738nicotinate-nucleotide pyrophosphorylaseCOG0157H0.550.44
***bcp4***
108331-Cys peroxiredoxin (PrxQ4)COG1225O0.400.43
***cheY6***
10887response regulator (receiver domain), 2C-systemCOG0784T0.440.40
***intA9***
11275site-specific recombinase (fragment)ndnd0.250.5
***hsdS***
11311type I DNA restriction specificity protein (part)COG0732V0.370.45
***nrtP***
11808nitrate/nitrite antiporterCOG2223P0.160.15
***narB***
11809nitrate reductaseCOG0243C0.180.21
***cynB***
11877cyanate ABC-type transport, membrane comp.COG0600P0.290.18
***cynD***
11878cyanate ABC-type transport, ATP-binding comp.COG1116P0.530.18
***cynX***
11879response regulator receiver domain proteinCOG1513S0.42019
***cynS***
11880cyanaseCOG1513P0.230.15
***cobA***
11943uroporphyrinogen-III C-methyltransferaseCOG0007H0.200.40
***nirA***
11944ferredoxin-nitrite reductaseCOG0155P0.340.28
***msrA1***
20193methionine sulfoxide reductaseCOG0225O0.440.43
***fmdA***
20218formamidase (formamide amidohydrolase)COG2421C0.470.31
***intB2***
20252site-specific recombinase (fragment)ndnd0.280.30
***msrPb***
30294methionine sulfoxide reductase subunit (part 2)COG2041R0.270.38
***msrPa***
30295methionine sulfoxide reductase subunit (part 1)COG2041R0.230.29
***hypA1***
40490hydrogenase expression/formation proteinCOG0375R0.320.33
***ndhD2***
40540NAD(P)H-quinone oxidoreductase chain 4COG1008C0.500.31
***murG***
40561N-acetylglucosaminyl transferaseCOG0707M0.430.37
***nrtD***
40618nitrate ABC-type transport, ATP-binding comp.COG1116P0.380.28
***nrtC***
40619nitrate import ATP-binding protein(b)P0.430.44
***nrtB***
40620nitrate ABC-type transport, permease comp.(a)P, L0.450.32
***nrtA***
40621nitrate ABC-type transport, periplasmic comp.COG0715P0.390.32
***banIR***
40641type II restriction enzyme BanICOG3587V0.420.45
***gmk***
40786guanylate kinaseCOG0194F0.410.39
***snaRb***
40882type II restriction enzyme SnaBI (part 2)COG3587V0.160.40
***snaRa***
40883type II restriction enzyme SnaBI (part 1)COG3587V0.210.33
***snaX***
40884R-M system control protein (prototype C.SnaBI)COG1396K0.200.35
***pcrA***
41347ATP-dependent DNA helicaseCOG0210L0.450.50
***nthB***
60176nitrile hydratase β subunitndnd0.360.27
***glcD***
60706glycolate dehydrogenase FAD-linked subunitCOG0277C0.460.48
***nblA1***
61056phycobilisome degradation proteinndS0.130.43(abbreviations, colors, and selection criteria are as in Table 3).


The *phaP* gene (_10501) is about threefold upregulated in both strains P2 and P6 (Table 5) and encodes a phasin that regulates the formation of polyhydroxybutyrate (PHB) granules [75]. This gene is part of a *phaECP* unit (_10499 to _10501) with *phaP* and *phaEC* convergently transcribed. The *phaEC* pair of genes, encoding the heterodimeric PHB synthase, was not scored in this experiment as DEGs. To date, no other PHB biosynthesis genes were identified in the *L. indica* PCC 8005 genome. As PHB is an important carbon/energy storage material in cyanobacteria [76,77] and may play a role in *Limnospira* survival mechanisms for γ-irradiated cells it may be worthwhile to search for additional genes in the PCC 8005 genome involved in PHB synthesis and look up their expression profiles obtained in our experiment.

The *metE* gene is strongly induced by gamma irradiation in both P2 and P6, with FC values of 7.6 and 4.9, respectively. Its gene product, homocysteine methyltransferase (also known as “methionine synthase”), catalyzes the formation of methionine from homocysteine thus providing, next to the biosynthesis of cysteine from serine, a second route of sulfur assimilation via protein synthesis and recycling. Both cysteine and methionine have critical roles in protein structure and function. While cysteine residues are involved in protein tertiary structure, protein–protein interaction, redox signaling, metal ion binding, and thiol-mediated antioxidant activities (for instance in thioredoxins) [78], methionine has a predominant role in protein initiation (in prokaryotes via the N-formyl methionine derivative) but is also deployed as an endogenous (intraproteinic) antioxidant [79,80]. Oxidized methionines (in the form of methionine sulfoxide or MetSO) originating from ROS attacks are repaired back to the original methionine by methionine sulfoxide reductase (MSR) so that they can take up again their ROS scavenging function in a catalytic cycle of oxidation and reduction [81]. In *L. indica* sp. PCC 8005 this important protein-repair enzyme is encoded by the *msrA1* gene (_20193) which is more than twofold repressed in gamma-irradiated P2 and in P6 cells (Table 5). The PCC 8005 genome harbours a second gene for methionine sulfoxide reductase (_11900) but this gene was annotated as being a fragment (fCDS) and, at first, was given little attention. Thus, while *L. indica* seems to step up the production of methionine upon exposure to ionizing radiation via the MetE biosynthesis pathway, perhaps providing excess methionine for the synthesis of anti-oxidant peptides, proteins, or enzymes, the ROS–methionine scavenging cycle might be disrupted by diminished MrsA levels. This to us makes little sense as we would expect that during oxidative stress MSR levels would be at least maintained or perhaps even induced. For that reason we turned our attention back to the presence of the second MSR gene (_11900) and found from the literature that cyanobacteria generally possess two genes encoding this enzyme, in addition to a third gene *msrB* [82]—consequently, we named gene _11900 as *msrA2* and gene _61123 as *msrB*. The A and B types of MSR display an absolute specificity towards the S- and R-MetSO isomeric forms, respectively, but do not share any similarity in sequence or structure. Both types are essential to reduce MetSO since oxidation of Met leads to a mixture of isomers. The MsrA1 and MsrA2 enzymes of *L. indica* sp. PCC 8005 share 45% aa sequence identity but they differ in length, i.e., 219 aa and 143 aa, respectively (for which reason the *msrA2* gene was probably considered a gene fragment in the MaGe annotation platform). As *msrA1* in our experiment is repressed in IR-exposed cells, and MSR-activity seems crucial during oxidative stress (i.e., due to gamma irradiation), we think that the *msrA2* gene product should be considered an active enzyme at least guaranteeing a basal level of intraproteinic MetSO-Met recycling. Importantly, neither *msrA2* nor *msrB* was regulated in P2 or P6 (Appendix A)—and none of the MSR encoding genes were regulated in ^60^Co-gamma irradiation studies on *L. indica* sp. PCC 8005 [11,12]. The reasons and mechanisms for *msrA1* shutdown upon SNF-gamma-irradiation in our experiment remain unknown for now.

Four chemotaxis-related genes *cheA1* (_60577), *cheB1* (_60572), *cheC1* (_60571), and *cheW1* (_60576) were also induced in both P2 and P6 (Table 5). These four genes are organized in two pairs and each pair is separated from each other by three genes: *cheR1*, encoding a chemotaxis methyl transferase (not scored as a DEG), gene _60574, encoding a 1091 aa large HEAT-repeat sensory protein (not scored as a DEG), and gene _60575, a chemotaxis related protein of undefined function (induced in strain P6, with an FC = 2.6 and an FDR = 0.000, but unregulated in strain P2 (Appendix A). A *cheY1* gene (_60578), encoding a two-component regulator, is located at the far end of this entire cluster. Although this latter gene was not scored as a DEG in P2 according to our strict criteria (and hence is listed in Table 4), it still had an acceptable FC of 1.98 with FDR = 0.008; in P6, FC and FDR were 3.5 and 0.000, respectively (Appendix A and Table 5). The activation of chemotaxis enzymes makes full sense for motile cyanobactaria such as *Arthrospira/Limnospira* who upon excesses of photonic energy move away out of danger while seeking out extra nutrients for adaptation and survival.

The chaperonin gene pair *groSL1* (Table 5) and particularly their massive induction in strain P6 have already been discussed extensively in the previous Section 3.2.2. Other induced genes in both strains P2 and P6 (Table 5) were involved in electron transport (*ubiA1*; FC ~2.5), carbohydrate biosynthesis (*rmlA*; FC~2.5), protein synthesis (*dusA*; FC ~4–5), vitamin biosynthesis (*panE*; FC ~3–4*, folE1*; FC ~5), pyrimidine biosynthesis (*pyrD*; FC ~2.2) and protein folding (*ppiC*; FC ~2.5), with seemingly no direct relevance to cyanobacterial responses towards ionizing radiation or oxidative stress except that all were involved, in one way or another, in the stimulation or re-direction of cellular resources.

The shutdown of the *stpA* gene in both P2 and P6 irradiated cells (Table 5) deserves a few words. It was named after *the stpA* (*slr0746*) gene of *Synechocystis* sp. PCC 6803 (the StpA proteins of PCC 6803 and PCC 8005 are 61% identical in aa sequence) where it was identified generically as a “salt tolerance protein” whose expression was enhanced at NaCl concentrations of 170 mM or above [83]. A few years later it was shown that the *Synechocystis* sp. PCC 6803 *stpA* gene actually coded for a glucosylglycerol-phosphate phosphatase (GGPP) [84], glucosyl-glycerol (GG) being a common compatible solute (osmoprotectant) of cyanobacteria. Seeing *stpA* being repressed we became interested in this gene because another solute, trehalose, appears to play an important role in the cellular protection of microorganisms against a variety of abiotic stresses including ionizing radiation [85,86] and we thought that perhaps GG synthesis was switched off in favor of trehalose production as we noted in previous irradiation experiments in *L. indica* PCC 8005 that gene expression for trehalose synthesis via maltose (TreS pathway) or dextrine (TreYZ pathway) were 70 to 300% enhanced in cells when exposed to high doses (800 Gy–1600 Gy–3200 Gy) of ^60^Co-gamma irradiation [11]. Additionally, we recently noted remarkable differences in trehalose content between P2 and P6 ^60^Co-gamma-irradiated cells [13]. Surprisingly, in our current experiment, neither *treS* (_41060) nor *treYZ* (_61152/3) was regulated in γ-irradiated P2 or P6 cells (i.e., at 3200 Gy of SNF γ-radiation).

The *bcp4* gene is approximately 2.5-fold repressed in both P2 and P6 strains (Table 5). This gene was annotated by MaGe as coding for a “bacterioferritin comigratory protein” but actually encodes a 1-Cys peroxiredoxin (i.e., holding only the peroxidatic Cys residue) based on the high aa sequence similarity to PerQ proteins of *Synechococcus* elongatus PCC 7942, *Synechocystis* sp. PCC 6803, and *Anabaena* sp. PCC 7120 [87]. Such peroxiredoxins have the general task to detoxify H_2_O_2_ from the cell [88], and it is surprising that this peroxiredoxin gene is switched off in response to ionizing radiation which is bound to produce ROS including H_2_O_2_. However, the PCC 8005 genome possesses three other *bcp* genes (however unregulated in our study) encoding two 2-Cys peroxiredoxins (*bcp1* and *bcp2*) (holding one peroxidatic Cys residue and one resolving Cys residue) and one atypical 2-Cys peroxiredoxin (*bcp3*) (holding one peroxidatic Cys residue and one resolving Cys residue but located further apart from each other) thus providing ample redundancy in H_2_O_2_-detoxifying capacity. Nonetheless, it might be worthwhile to find out why and how *bcp4* expression is blocked (note that all four *bcp* genes have now been renamed in the MaGe database as *prxQ* correspondingly numbered 1 to 4).

The commonly repressed *msrA1* gene was discussed in the context of MetSO-Met recycling (see higher up in this section). Two genes _30294/5, both identified as fCDS in MaGe, appear to encode parts of a novel methionine sulfoxide reductase (MRS) and were also repressed in both strains P2 and P6. These two fCDS form together one gene in all other *Arthrospira/Limnospira* genomes thus may be the result of either a mutation or sequencing error. Amino acid sequence alignment with the *E. coli* MsrP protein (UniProtKB—P76342) learned that the _30294/5 pair corresponds well with respectively its carboxy and amino-terminal ends and hence for now we called these genes *msrPa* and *msrPb* until the question of one or two CDS has been resolved. The *E. coli* MrsP protein is capable of in vitro reducing N-acetyl-Met-O, a substrate mimicking protein-bound Met-O, implying a function in the repair of ROS-oxidized proteins [89]. The *E. coli* MsrP partner MsrQ, a heme-binding membrane protein, was not readily identified in the PCC 8005 proteome (using BLASTp with UniProtKB—P76343 as a query). Possibly, when *L. indica* PCC 8005 has to cope with prolonged radiation stress, the concerted action of MsrA1, MrsA2, MrsB and tentatively MsrPab suffices to keep pace with the required intraproteinic Met recycling from MetSO, even at lower *msrA1* expression, particularly when cellular Met levels are abundant since the Met biosynthesis gene *metE* is highly induced by γ-radiation in both P2 and P6. The exact reason why *msrA1* gene expression is repressed over twofold by γ-radiation in both P2 and P6 remains elusive but in fact, there might be a correlation between *metE* induction and *msrA1* repression in the sense that an excess of methionine in the cell may have a negative feedback effect specifically on *msrA1* transcription so that not γ-radiation but actually methionine abundancy is the immediate effector.

The *hypA1* gene (_40490) encoding a putative hydrogenase maturation factor was threefold repressed in both strain P2 and P6 (Table 5). This gene appears to be part of a cassette of six genes (_40486 to _40491) displaying a perfect synteny to a *Synechocystis* sp. PCC 8603 operon of six genes *sll1077* to *sll1082*, with high gene-to-gene similarity in length and aa sequence (55–85% aa identity). Besides _40489, named *hypB1* after MaGe predicts it as a second hydrogenase maturation factor, none of the four other *L. indica* genes were given a name (and hence escaped our attention as “unknowns”). Yet, the three genes downstream of *hypB1* were annotated as the three subunits of an ABC transport system while the gene preceding *hypA1* was annotated as an agmatinase (involved in arginine and proline metabolism). Because the *hypB1* expression is also regulated, i.e., repressed threefold in P6, but is not seen as a DEG in P2 (Table 4), we checked the expression profiles for the four unnamed genes. As it turns out, the agmatinase-encoding gene was threefold repressed in strain P6 (but not scored as a DEG in P2) while the three transport-related genes were repressed three to four-fold in P6 (but again not seen as DEGs in P2) (Appendix A). Thus, all six genes are downregulated in strain P6 but only *hypA1* is also repressed in strain P2. Perhaps *hypA1* and *hypB1* are actually not maturation factors for the HOX hydrogenase, which is encoded by a distantly located locus *hoxEFUYH* (_41294 to _41299), but instead for the agmatinase encoded in the same _40486 to _40491 locus. It is also not clear what is being taken up or exported by the ABC transporter encoded in this locus and what its structure might be. Given the lack of reliable and conclusive functional information on these six genes, it is difficult to assess their relevance in terms of the *L. indica* PCC 8005 response/resistance to ionizing radiation, but it illustrates well how genomic, ontological, and transcriptomic data can work together to improve our understanding of bacterial gene networks, or at least identify interesting loci for further research.

Cyanobacteria adjust the amount and composition of their light-harvesting pigments in response to environmental cues by the action of a small peptide (coined NblA) that acts as a proteolysis adaptor protein required for the disassembly and degradation of phycibillisomes [90]. This feedback mechanism basically prevents photoinhibitory damage in times of surplus excitation (e.g., continuous high-light conditions). It has been shown in *Anabaena* sp. PCC 7120 that homodimeric NblA interacts with ClpC, an HSP70 (ClpB) chaperone partner, guiding the ClpC-ClpB proteolytic complex to the phycobiliprotein disks in the rods of phycobilisomes [91]. More recently, a NblA1/NblA2 heterodimer made of the products of two *nblA* genes has been implicated in the degradation of *Synechocystis* sp. PCC 6803 phycobilisomes [92]. The *L. indica* PCC 8005 genome also contains two such genes: *nblaA1* (_61056) and *nblA2* (_61095), located about 50 kb apart, displaying ca. 40% aa identity to each other. Besides the fact that we do not know whether proteolytic degradation of the *Limnospira/Arthrospira* phycobilisome requires both *nblA* genes and whether this degradation is mediated by a homedimeric or heterodimeric NblA adaptor, previous ^60^Co-gamma irradiation experiments have shown that *nblA2* was twofold upregulated by application of a 527 Gy·h^−1^ dose rate [12] but was not regulated by ^60^Co-gamma irradiation when cells were exposed to the very high dose rate of 20 kGy·h^−1^ [11], while for *nblA1* no regulation was seen in either dose rate. In our experiment, with an SNF-gamma radiation dose rate of “only” 80 Gy·h^−1^, but over a longer period of exposure time, i.e., days and not hours or minutes, and in the presence of light, *nblA1* was downregulated in both strains P2 and P6 (Table 5) while *nblA2* was downregulated in the P2 strain only (Appendix A) but was not validated as a DEG via normalization. This may point to a strategy of stalling phycobilisome degradation in favour of light-harvesting for energy, i.e., by keeping the production of this key peptide, NblA, as low as possible.

Besides the ammonium, nitrate/nitrite, nitrile, and cyanate routes already discussed at the end of the previous Section 3.2.2, *L. indica* has yet another route in nitrogen metabolism and assimilation at its disposal, i.e., in the form of the *fmdA* gene (_20218) encoding a formamidase. This enzyme essentially frees up ammonia from organically stored nitrogen in the form of amides, most notably formamide. The 2.5–3.5 fold repression of *fmdA* is in line with the shutdown of nitrogen metabolic pathways (outlined in 3.2.2) and our previous observations on *L. indica* responses to IR [11,12]. All other commonly repressed nitrogen-related genes listed in Table 5 (*nrtP*, *narB*, *cynBDXS*, *nirA*, *nrtABCD*, *nthB*) have also been mentioned in the previous section. The remaining genes repressed in both P2 and P6 are involved in transport (*livG*), cofactor synthesis (*nadC*, *gmk*, *cobA*), signal sensory (*cheY6*), DNA replication (*pcrA*), carbon metabolism (*glcD*), electron transport (*ndhD2*), cell wall biogenesis (*murG*), and DNA restriction and modification (*yhdJ, hsdS*, *banIR*, *snaRab*, *snaX*).

#### 3.2.4. RNA Genes Regulated by γ-Radiation

Out of the 337 non-coding RNAs (ncRNA) identified in the *L. indica* PCC 8005 genome via the MaGe platform, 58 were found to be regulated using the strict DEG selection criteria −1 ≥log_2_FC ≥ 1 and FDR ≤ 0.05 of which 26 were up- and 32 were downregulated (Table 2 and Appendix A). Of those, 32 genes transcribing non-coding RNA were withheld after normalization, i.e., 10 induced and 22 repressed (Table 6, Table 7 and Table 8). We do not include the 14 group I/II introns and HEARO RNAs in our discussion because we consider them as post-splicing, post-mobility intron remnants. Nonetheless, future analysis, e.g., in regard to their precise location should be undertaken to check whether their presence might be affecting the function of host genes or, in the case of intergenic location, nearby genes.


microorganisms-09-01626-t006_Table 6Table 6Irradiation-induced and repressed RNA genes in P2 but not in P6.GeneSizeType/FunctionRfam-IDFC
**RNA153**
65grp II intronRF000292.20
**RNA220**
78grp II intronRF000294.35
**RNA248**
78grp II intronRF000295.60
**RNA273**
78grp II intronRF000295.04
**RNA285**
65grp II intronRF000292.31
**RNA105**
78grp II intronRF000290.37
**tRNA15**
74Pro tRNARF000050.33
**tRNA17**
73Trp tRNARF000050.27
**tRNA35**
73Phe tRNARF000050.34
**tRNA41**
75Thr tRNARF000050.44(abbreviations, colors, and selection criteria are as in Table 3).
microorganisms-09-01626-t007_Table 7Table 7Irradiation-induced and repressed RNA genes in P6 but not in P2.GeneSizeType/FunctionRfam-IDFC
**RNA67**
78grp II intronRF000292.20
**tRNA38**
73Arg tRNARF000050.45
**RNA90**
78grp II intronRF000290.36
**RNA116**
237grp I intronRF000280.39
**RNA124**
78grp II intronRF000290.03
**RNA134**
79grp II intronRF000290.24(abbreviations, colors, and selection criteria are as in Table 3).


Transfer RNAs (tRNAs) are indispensable molecules in the translational machinery by which the genetic information in the mRNA, through 61 different triplets (codons), is decoded into a peptide or protein. The *L. indica* PCC 8005 genome possesses a total of 42 tRNAs recognizing these 61 codons. Strikingly, 16 of these RNAs are repressed in gamma-irradiated cells (four in P2 only, one in P6 only, and 11 in both P2 and P6) while none are induced (Table 6, Table 7 and Table 8). Stability and modification of tRNAs and the balance of tRNA supply, both in quantity as well as in composition, are determining factors in stress-dedicated protein synthesis [93,94,95,96]. With a number of pathways involved in central metabolism and amino acid synthesis diminished upon exposure to gamma radiation, *L. indica* PCC 8005 appears to rearrange its tRNA pool to address priority changes of protein synthesis. It may also be possible that it attempts to limit or avoid the production of proteins holding certain aa residues that are particularly prone to ROS attack or redox-mediated modification, e.g., Trp, Tyr, Phe, and His [97,98]. To find out the exact reasons for the drastic repression of tRNAs in gamma-irradiated *L. indica cells*, a thorough analysis of the concerned tRNAs (i.e., what anticodon is affected, what is the role of the resulting residue in proteins, etc.) and genetic network analysis (stringent response, tRNA modification, tRNA stability, …) are required.


microorganisms-09-01626-t008_Table 8Table 8Irradiation-induced and repressed RNA genes common to P2 and P6.GeneSizeType/FunctionRfam-IDFC (P2 and P6)
**RNA2**
78grp II intronRF000292.342.41
**RNA68**
79grp II intronRF000292.382.61
**RNA269**
146cobalaminRF001742.812.12
**RNA280**
65grp II intronRF000292.112.39
**tRNA11**
74Arg tRNARF000050.220.27
**tRNA13**
77Val tRNARF000050.180.25
**tRNA14**
72Gln tRNARF000050.320.40
**tRNA18**
82Tyr tRNARF000050.390.30
**tRNA19**
72Thr tRNARF000050.370.30
**tRNA26**
83Leu tRNARF000050.240.35
**tRNA27**
72Val tRNARF000050.350.34
**tRNA31**
71Gly tRNARF000050.360.31
**tRNA32**
72Gly tRNARF000050.290.16
**tRNA37**
73His tRNARF000050.190.42
**tRNA39**
90Ser tRNARF000050.480.33
**RNA182**
149ykkC-yxkDRF004420.420.29(abbreviations, colors, and selection criteria are as in Table 3).


Two RNA genes regulated by SNF-gamma irradiation in both P2 and P6 strains (Table 8) are worth mentioning:

(i) RNA269 representing a cobalamin riboswitch which is located immediately upstream of the *metE* (_60603) gene and was upregulated 2–3 fold by γ-irradiation in both strains (Table 8), as is the *metE* gene itself which was upregulated 5–8 times (Table 5; discussed in Section 3.2.3). Although RNA269 is 337 bp away from the *metE* start, it is likely that it is part of the *metE* 5′ untranslated transcribed region (5′UTR). Riboswitches are elements that exert regulatory control in a cis-fashion, most often over the transcript in which they are embedded, via two secondary-structure domains, the receptor domain binding a small effector molecule (which can be a metabolite, a signaling molecule, or an ion) and a regulatory switching domain that interfaces with either the transcriptional or translational machinery (or both) thereby directly affecting expression [99]. The effector specificity is usually very high and is determined by riboswitch local RNA sequence and structure. RNA269 was identified by MaGe based on its similarity (Expect value of 2.2 × 10^−11^) with cobalamin riboswitches where the effector is (one of the chemical forms of) cobalamin. However, cobalamin riboswitches are almost exclusively found in the 5′ UTRs of cobalamin biosynthesis genes. The location of RNA269, likely being part of the *metE* 5′ leader sequence, suggests that the effector would be actually S-adenosylmethionine (SAM), as is the case for the *B. subtilus metE* gene and its riboswitch [100]. Detailed sequence analysis is required to determine the presence of an “S-box” rather than a “B12-box” (these boxes are sequence elements that are indicative for the effector to be bound—[101]) while it would also be interesting to study RNA269 mutants under various growth conditions as methionine and sulfur are all-important in cyanobacteria and perhaps particularly so in the resistance to ionizing radiation (IR) (see Section 3.2.1 and Section 3.2.2)

(ii) RNA182 forming a 149 nt RNA species with the resemblance in sequence and structure to the ykkC-yxkD leader, a conserved RNA structure found upstream of the *ykkC* and *yxkD* genes in *Bacillus subtilis* and related genes in other bacteria and characterized as guanidine-sensing riboswitches that function to switch on efflux pumps and detoxification systems in response to perilous conditions [102,103]. The RNA182 gene is located immediately upstream of the _40491 gene which codes for an agmatinase (agmatine ureohydrolase) responsible for the hydrolysis of agmatine to urea and putrescine and which is part of a six-gene cassette holding the *hypA1* and *hypB1* genes (Table 4 and Table 5) encoding two enzyme accessory proteins—as well as three genes encoding an ABC transporter (genes _40486 to _40491). These genes were discussed earlier in this section: the agmatinase-encoding gene was 3-fold repressed in strain P6 but not scored as a DEG in P2 while the three transport-related genes were repressed 3–4 fold in P6 but again not seen as DEGs in P2 (Appendix A). The RNA182 gene is clearly repressed in both gamma-irradiated P2 and P6 cells (Table 8; FC = 2.4 to 3.5) and its position suggests it is an integral part of the 5’ UTR of the _40491 agmatinase gene. The aforementioned six-gene cassette does not seem to be very common in *Limnospira/Arthrospira* with currently—out of seven genomes in the MaGe system—only the *Arthrospira* sp. TJSD091 genome also displaying this cassette. However, the complete cassette is also present in *Synechocystis* sp. PCC 6803 where it also bears the ykkC-yxkD leader in the 5′ UTR of the first gene (*sll1077*) [104]. To date, no further information is available on this locus.

Because small non-coding RNAs do not impose any metabolic burden on host cells yet are often instrumental, even at minute changes in their own expression, in global or specific gene regulation in response to cyanobacterial stress [105,106,107], a closer look is warranted regardless of the strict DEG selection criteria or the validation check through normalization as we applied in this study for the protein-encoding genes. Four genes are of particular interest: RNA98, seen as three-fold induced in only P2, and RNA162, RNA200, and RNA242, seen as twofold repressed only in P6 (Appendix A). The RNA98 gene encodes so-called iron stress repressed RNA or IsrR. This is actually an anti-sense RNA (asRNA) transcribed from the opposite strand of the *isiA* gene (see Section 3.2.1) and is able to bind, under sufficient iron conditions, to the central part of *isiA* mRNA forming a duplex RNA target for enzymatic degradation, thus modulating the expression of the IsiA protein [108]. The fact that *isrR* and *isiA* are co-induced (in P2 but not P6) may seem weird but in fact, this is because their RNAs are degraded together at different rates rendering intricate stoichiometric concentrations of IsrR asRNA and *isiA* mRNA achieving a fine balance of *isiA* gene activation and inactivation, with an initial delay during an early stage of stress and a fast decrease at the end of stress (releasing chl *a* for immediate use) quickly followed by the onset of recovery under normal growth conditions [109]. The RNA162 gene encodes a small RNA (sRNA) only 57 nt long and resembles the nitrogen stress-induced RNA1 (NsiR1) detected in a number of cyanobacteria where it is expressed very early and transiently at the onset of dwindling nitrogen levels [110]. Its expression requires NtcA but also HetR, a heterocyst-specific transcriptional regulator and it has been suggested that NsiR1 can be used as an early marker for cell differentiation in cyanobacterial filaments [111]. Although *Limnospira/Arthrospira* do not form heterocysts nor fix N_2_, all their genomes carry a *hetR-patS* locus, PatS being a diffusible inhibitor of heterocyst formation regulating spacing of heterocysts along the length of filaments, and HetR levels increased in *Arthrospira platensis* following combined-nitrogen removal [112]. These authors also hinted at the presence of “pigment-rich cells” visible by red fluorescence and placed regularly along the filament and hypothesized that these cells ensured the survival of at least some of the cells under adverse conditions. It is thus feasible that a shutdown of nitrogen assimilation and metabolism as part of a larger radiation response evokes NsiR1 expression, and that this occurs specifically in P6 but not in P2 because of the P6-specific repression of *glnA*, *glnB*, *ntcB*, *amt1*, etc.; see previous sections). Actually, NsiR1 levels in irradiated P2 rather point to induction of expression, with an FC = 1.88 and an FDR = 0.113) (Appendix A). Possibly, the shutdown of N_2_ assimilation and/or metabolism in P2 occurs at a different pace or to a different extent, once again emphasizing the idea that strains P2 and P6 follow their own agenda in their response to the prolonged exposure to (SNF) gamma radiation. The RNA200 product belongs to the Yfr2 family of non-coding RNAs identified in almost all studied species of cyanobacteria and are characterized by a so-called Cyano-1 RNA sequence motif [113]. The majority of Yfr2 genes appear as individual transcriptional units, possessing their own promoter. In *L. indica* PCC 8005, RNA200 lays immediately upstream of gene _40989 which encodes a conserved hypothetical membrane protein (possibly a cytochrome B but the analysis was inconclusive); added note: this _40989 gene is about twofold induced in gamma-irradiated P6 cells but not in irradiated P2 cells (FC = 2.1, FDR = 0.008) (Appendix A). The biological function of Yfr2 RNAs is still enigmatic but they seem to play a crucial and global role in carbon- and nitrogen-related primary metabolism, photosynthesis, and respiration through the interaction with other ncRNAs and asRNAs or by targeting certain transcriptional regulators [114]. The RNA242 gene (*ssaA*) encodes a 185 nt long 6S RNA whose secondary structure resembles an open promoter complex through which it binds to RNA polymerase and acts as a regulator of sigma 70-dependent transcription in many prokaryotes [115,116]. In *Synechocystis* sp. PCC 6803, the *ssaA* ncRNA has an integral role in the cellular response to changes in nitrogen availability by facilitating the switch from group 2 sigma factors SigB-, SigC-, and SigE-dependent transcription to SigA-dependent transcription [117].

#### 3.2.5. Genes with Unknown Function at Least Fivefold Regulated by γ-Irradiation

Based on simple COUNTIF operations in Appendix A, 60 genes in P2 and 33 genes in P6 are induced by γ-radiation with log_2_FC ≥ 2.322 (FC ≥ 5.00), of which 21 are common to both P2 and P6, while 40 genes in P2 and 20 genes in P6 are repressed by γ-radiation with log_2_FC ≤ −2.322 (FC ≥ 5.00), of which 6 are common to P2 and P6. As before, in this count, only genes with an FDR ≤ 0.05 are considered. This count obviously also includes all named genes with FC ≥ 5 and FDR ≤ 0.05 (Appendix A).

We provide two sheets in Appendix A displaying fivefold induced (*n* = 22) and fivefold repressed (*n* = 11) unnamed genes (i.e., after batch normalization), always with FDR < 0.05. For each of the unnamed genes, we checked functional evidence in the MaGe annotation platform. As was to be expected, the majority of these unnamed genes can only be described as hypothetical or conserved hypothetical proteins due to the lack of any evidence on possible function. In only very few instances MaGe detected a known protein domain or suggested a putative function (this general picture already emerged for all the unnamed genes in the gene lists obtained by COUNTIF operations on Appendix A, i.e., prior to batch normalization).

Researchers interested in highly regulated unnamed genes of this study can retrieve gene lists from Appendix A using custom FC and FDR cutoff values, and subsequently look up those genes in MaGe for additional functional information, or download data from MaGe and perform dedicated bioinformatic analyses. Even with little functional information on a particular gene, if this gene is immediately adjacent to a named gene with a known function considered as a DEG under our criteria (e.g., FC ≥ 2, FDR ≤ 0.05) this may be a clue for further investigation. Likewise, clusters of genes that are co-regulated by γ-radiation may be of special interest, even if none of those genes have a predicted function; an example is the five *arh* genes listed in Table 5 (ARTHROv5_10467 to _10471), each of which was induced at least 8-fold by γ-radiation in both P2 and P6. In fact, these genes were included in Table 5 as they were previously given a name in MaGe [11,12] (although we now renamed these genes in the order *arhA* to *arhE* to be in line with transcriptional direction).

#### 3.2.6. Association of P2 and P6 Expression Patterns with Their Respective Genotype

In our previous study [13] we observed that the *L. indica* PCC 8005 morphotypes P2 and P6 behaved differently in terms of growth and buoyancy and also displayed after exposure to γ-radiation distinct differences in antioxidant capacity, pigment content, and trehalose levels. In the same study, the whole-genome comparison revealed a difference of 168 SNPs, 48 indels, and four large insertions affecting in total 41 coding regions (CDS) across both genomes of which only nine could be assigned a function. Of those nine CDS, four were severely affected by a frameshift or large insertion: _10705 and _11989 in P2 and _60747 and _30483 in P6 (Table 9). The other five CDS harbored single or multiple amino acid substitutions with unclear functional outcomes. In addition, a total of 56 SNPs or indels were detected in 34 intergenic regions across both strains [13]. The vast majority of the affected intergenic regions separate genes that encode proteins of unknown function, gpII introns, or transposases (or fragments thereof) while many SNPs or indels were located over 250 nt away from the nearest downstream gene or located in between two converging genes, with less or no direct impact on expression. The remaining six genes whose expression might have been affected by an upstream SNP or indel (all detected in the genome of the strain P6) are _11992 (*ycf4*, encoding a PSI assembly protein), _11993 (*psbD*, encoding the D2 protein of the photosynthesis PSII complex), _60118 (encoding a DNA-[cytosine-5-]-methyl transferase), _60128 (encoding a fibronectin-binding-A-like protein), _60723 (encoding a signal transduction histidine kinase), and _61273 (encoding part of a tetratricopeptide TPR_2 repeat protein).


microorganisms-09-01626-t009_Table 9Table 9Affected CDS with known function in strains P2 and P6 based on genome data (taken from [13]).MaGe-IDFunctionSize (aa)Strain P2Strain P610196adenylosuccinate synthetase (PurA)446
C248G10705Ser/Thr protein kinase825E290fs
11989hemolysin-type Ca-binding protein1261
V592R, L596R, A597D + insPDGPDPEL12033gas vesicle protein (gvpC)151
K135D30483putative Ser/Thr protein phosphatase360
large insertion30654nitrilase/cyanide hydratase269L21F
41442putative diguanylate cyclase195G136R, T172A, C176R
60747signal transduction histidine kinase790
L443 *61039WD-40 repeat protein818A124GQ100K, T106R, E804GMaGe-ID, ARTHROv5 unique gene identifier of the MaGe Genomes Database (https://mage.genoscope.cns.fr/) [10], the creation of a stop codon (resulting in a truncated gene product) is denoted as an asterisk.


Of the above nine affected CDS with known function and six genes possibly affected in their upstream regulatory region, only _61039 (Table 9) and _61273 were differentially expressed before and after exposure to γ-radiation. Gene _61039 encoding a WD-40 repeat protein was about two-fold induced by γ-radiation in both P2 and P6 (with an FC of 1.97 and 2.11, and an FDR of 0.02 and 0.01, respectively) (Appendix A), while gene _61273 encoding a TPR repeat protein was repressed fourfold in P2 (FC = 3.84, FDR = 0.00) (Appendix A) but not considered a DEG in P6, neither by FC nor by FDR. The function of these repeat proteins in cyanobacteria is not well understood but it is thought that they play an important role in protein-protein interactions, protein complex formation and stabilization, and the interaction with macromolecules in a wide variety of cellular superstructures and processes [118,119].

With the current lack of functional information for the majority of *L. indica* PCC 8005 genes, it is for now not possible to perform any meaningful associative analysis between the genotypes of the P2 and P6 substrains and their different metabolic and physiological responses to IR. Additionally, it is important to keep in mind that genotypic changes between P2 and P6, whether in gene coding or regulatory regions, may cause cascade-driven and pleiotrophic effects that cannot be easily traced and even may act in a combinatory fashion and/or on a global scale. Clearly, special efforts are needed to improve the functional annotation of the *L. indica* PCC 8005 genome/proteome. However, equally important is the development of a genetic system allowing site-directed mutational analysis and the isolation and genotypic characterization of naturally occurring IR sensitive *L. indica* strains—which we have not encountered yet over the past several years of testing isolates from various sources and geographical locations, although a variation in IR resistance does exist for *Limnospira* and *Arthrospira* strains in the range of 2–5 kGy (unpublished results).

## 4. Conclusions

Although the cellular routes used by the *L. indica* PCC 8005 substrains P2 and P6 to cope with ionizing radiation (under the conditions applied, i.e., during one lifecycle under 45 μE·m^−2^·s^−1^ continuous light and 80 Gy·h^−1^ SNF γ-irradiation) overlap each other to a large extent—as exhibited by the many co-regulated genes across the two, such as shutting down central metabolism, amino acid biosynthesis, and photosynthesis in favor of repair and ROS detoxification—each strain displayed a preference of priorities, most probably brought about by their slightly different genetic backgrounds.

In order to narrow down the number of genes for analysis, we focused on those transcribing non-coding RNA or that had been given a name (i.e., *glnA*) by the MaGe annotation system [10] (started in 2010 and still ongoing to date at a slow pace—the gene name is a strong indication on the gene’s function and a powerful “handle” to manage and preselect expression data). However, unnamed genes with informative lines of functional evidence in their MaGe “gene cards” were missed. In retrospect, it might have been better to follow a slightly different route by not relying solely on the gene name but also taking into account the gene product description provided by MaGe. That said, this may only shift the problem as this information then needs to be qualified. We need to look into this but it is obvious that any post-annotation analysis primarily depends on the actual quality of the annotation. Thus, while name-giving strongly indicates the known function, one cannot blindly assume that all gene names are correct nor unique. In many cases of duplicate genes (*petJ*, *nblA*, *pmbA*, *cnr*, …) we had to resort to indexation (*petJ2*, *nblA1*, etc.) and in several cases of gene fragmentation (e.g., *hsdR1*, *dnaK3*), to subindexation (a, b, c, …). This was a time-consuming interactive process during which structural and functional annotation in MaGe was scrutinized and improved on a case-by-case basis and, where necessary, genes were renamed (e.g., *bcp4* to *perQ4*) or newly identified genes were given for the first time a new name (e.g., *gifA* and *gifB*).

After detailed gene analysis, we postulate that P2 succeeds in swiftly shutting down pathways irrelevant for basic metabolism and adjusting cellular activities in terms of DNA replication, cell division, and amino acid and nucleotide synthesis, hence conserving important amounts of energy to remodel transcriptional and post-transcriptional mechanisms allowing the redirection of resources towards cellular survival. The P6 strain is not equally successful in this response so it seems, and as a consequence, needs to focus on emergency measures involving enhanced DNA and protein repair and overall damage control. Some transcriptional regulators and sigma factors as well as some crucial regulatory ncRNAs are differently expressed in P2 versus P6 while also DNA methylation and circadian rhythm likely differ across the two strains. These factors govern multiple target genes whose expression, in turn, may define the production of other cell components, and so forth. It is thus expected that alterations in such regulatory cascades and networks have a decisive, cumulative effect on cellular function at the molecular level.

Of the many genes regulated by SNF γ-radiation in either P2 or P6 but not in both, four genes, in our opinion, deserve immediate further study in a P2 to P6 comparison: *groL2* (only induced in P6; FC = 11.1), *isiA* (only induced in P2; FC = 2.3), and the two genes, *rfpX*, and *hliA* (repressed in P2 only; FC = 2 to 5). We highlight the *groL2* gene because we think the GroL2 chaperonin might be specifically produced in P6 to counter radiation-induced protein damage and may interact with many radiation-damaged proteins, *isiA* because it encodes a protein with a dual role of protecting PSII from excesses of excitation energy and storing large amounts of chlorophyll for immediate post-irradiation use, and *hliA* and *rfpX* because they encode proteins instrumental for an optimal photosynthetic apparatus. These studies should entail the design of gene-specific primers and gene expression measurements by RT-qPCR in response to various conditions of ionizing radiation (or oxidative stress) as well as detailed proteomic studies with a focus on certain cellular pathways.

Other genes that similarly warrant immediate attention are *glnA* and *ntcA*. The *glnA* gene because of its pivotal role in C- and N-metabolism and because it was firmly repressed in previous gamma irradiation experiments while in our experiment it was only repressed in strain P6 and not in P2, and the *ntcA* gene because it encodes a global transcriptional regulator of many target genes (*glnA, gifA, gifB, glnB, nirA, narB, nrtcABCD*, *amt1*, *metX, rbcL, rbcS, cynABDS, nblA, pstS, sigD, folE, hetR*, …) involved in different cellular processes and because it is not regulated by radiation in our experiment, in contrast to previous irradiation experiments [11,12] when *ntcA* was repressed.

In this study, we set out to unravel the radioresistance mechanisms in *L. indica* PCC 8005 by relating the differences in RNA expression patterns between two of its sibling strains P2 and P6 to their different responses to SNF γ-radiation. This proved to be a difficult exercise because: (i) the stringent DEG selection criteria possibly masked interesting observations (a twofold change may not ideally balance data complexity with gains of insight), and (ii) a lot of information was missing since the majority of regulated genes (~85%) in this study were unnamed, encoding proteins of unknown function, and hence disregarded. A better view will be obtained on radioresistance in *L. indica* using our data if this percentage of “unknowns” can be brought down and hence renewed efforts should focus on an improved functional annotation of the *L. indica* PCC 8005 genome data.

## Figures and Tables

**Figure 1 microorganisms-09-01626-f001:**
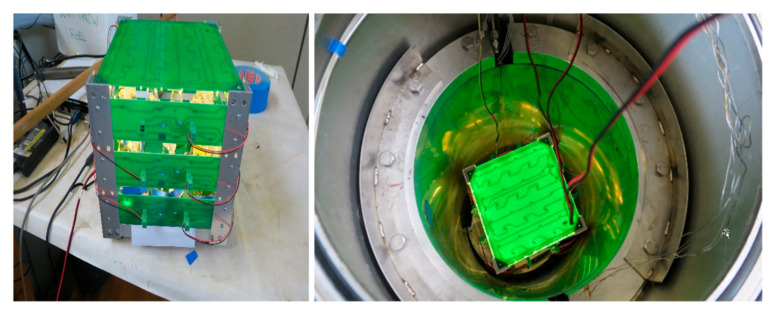
LED light tower assembled from panels connected in parallel, with culture flasks receiving continuous white light (**left**) and its positioning inside the GEUSE II vessel for exposure of *Limnospira indica* PCC 8005-P2 and -P6 cultures to spent nuclear fuel (SNF) gamma rays (**right**).

**Figure 2 microorganisms-09-01626-f002:**
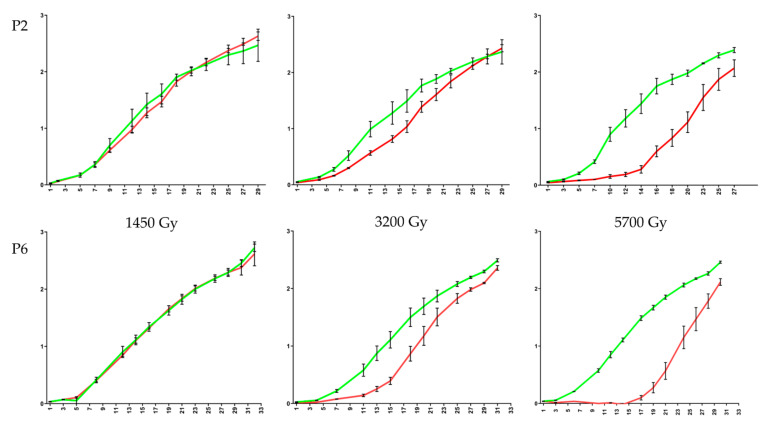
Growth curves of *L. indica* PCC 8005 subtypes P2 and P6 plotted as optical density at 750 nm (OD_750_) (*y*-axis) versus time in days (*x*-axis). Red curves represent gamma-irradiated cultures while green curves represent non-irradiated control cultures. Samples for post-irradiation outgrowth were taken in triplicate at three timepoints amounting to cumulative doses of 1450, 3200, and 5700 Gy. At the same time points triplicate samples were taken from non-irradiated control cultures. Data represent the mean of three independent biological replicates, and error bars present the standard error of the mean (SEM).

**Table 1 microorganisms-09-01626-t001:** Transcriptomic studies on IR-exposed *L. indica* PCC 8005.

Source	Rate (Gy·h^−1^)	Exposure (Max Dose) ^d^	IR Doses (Gy) ^e^	Light	Technology	Reference
^60^Co ^a^	20,000	9.6 min	800–1600–3200	no	MA—tiling ^g^	[11]
^60^Co ^b^	527	11.5 h	3200–5000	no	MA—tiling ^g^	[12]
SNF ^c^	80	3 d	1450–3200–5700	LED ^f^	RNA-Seq ^h^	This work

^a^ gamma radiation from the BRIGITTE facility at SCK•CEN; ^b^ gamma radiation from the RITA facility at SCK•CEN; ^c^ SNF: gamma radiation from rods of Spent Nuclear Fuel (SNF), GEUSE II facility at SCK•CEN); ^d^ prechosen sampling times determined the cumulative doses during the experiment; ^e^ approximate values (see methods); ^f^ warm white LED light at 45 μE.m^−2^·s^−1^ (see Methods for details); ^g^ tilling microarray (MA) analysis by Roche NimbleGen, USA; ^h^ RNA-Seq performed by NXTGNT, Belgium; all transcriptional analyses were based on genome version v5 (ARTHRO_v5) of 15 February 2014, Genbank accession number GCA_000973065 [9].

**Table 2 microorganisms-09-01626-t002:** Breakdown of gamma-irradiation regulated genes unique to P2 or P6 or common to both.

	Induced (F/R)	Repressed (F/R)	Total
**P2**	336 (42/21)	551 (77/22)	887
**P6**	398 (55/9)	268 (59/22)	666
**common**	208 (28/4)	144 (34/12)	352
**across P2 and P6 (1)**	69/26	102/32	229 (a)
**across P2 and P6 (2)**	46/10	52/22	130 (b)

Base numbers were calculated from Appendix A. F, number of genes with predicted function; R, number of genes transribing non-coding RNA. Calculations of induced and repressed genes across the two strains in (1) and (2) take into account the common genes, i.e., 42 + 55 − 28 = 69 and are before and after verification by batch normalization, respectively. The final 130 genes are further broken down in Table 3, Table 4, Table 5, Table 6, Table 7 and Table 8.

## Data Availability

The quantitative transcriptomic (RNA-seq) data including the raw sequencing reads, individual sample gene expression levels, and normalized abundance gene expression levels have been deposited in the National Center for Biotechnology Information (NCBI) Gene Expression Omnibus database under the accession number GSE175921. These data can be accessed directly via https://www.ncbi.nlm.nih.gov/geo/query/acc.cgi?acc=GSE175921 using a secure token that can be requested from the correcponding author. Two months upon acceptance of publication, access will be made public and no token will be further needed.

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
