# Peer review of "Genetic Responses of Metabolically Active Limnospira indica Strain PCC 8005 Exposed to γ-Radiation during Its Lifecycle"

_microorganisms, 2021, doi:10.3390/microorganisms9081626_

Round 1
Reviewer 1 Report
Yadav et al,
Genetic responses of metabolically active Limnospira indica strain PCC 8005 exposed to g-radiation during its lifecycle
This manuscript describes RNA-seq analyses of two morphotypes of Limnospira indica strain PCC 8005 after exposure to chronic gamma radiation. It is a long but well-written manuscript. The methods, data analysis and results are, mostly, clearly described and discussed. The results show amongst others that the genetic response towards a stress may differ to some extent in two derivatives of the same strain, as observed here for L. indica strain PCC 8005.
I have a few comments and suggestions.
The name/word “cyanobacteria” is mentioned for the first time on page 8 and not before. I suggest to mention in the Introduction (and perhaps also in the Abstract?) that Limnospira belongs to the Cyanobacteria.
At various places in the manuscript, the authors mention an amount of genes that were found differentially expressed, but I don’t think that the total amount of predicted genes is mentioned. It would be good to mention somewhere the total amount of predicted genes, as well as the genome size of PCC 8005 (for example near line 52, or elsewhere if more appropriate).
3.
The authors refer to the MaGe platform for the most recent (v5) genome annotation of L. indica strain PCC 8005, but I did not find this genome or the genes (using the MaGe-IDs as given in the tables or text) on this website. I also tried Arthrospira PCC 8005, but this seems to be v1 on MaGe. Is ARTHROv5 publicly available at MaGe? If not, will it be?
I did find L. indica PCC 8005 proteins at UniProt, but the locus tags at UniProt appear different compared to MaGe-IDs in the manuscript. For example, KaiB at UniProt is ARTHRO_490005 whereas it is ARTHROv5_60141 in this manuscript.
In Table S1, the translations of the protein-coding genes are given (with locus tag); that’s good because the reader can find proteins of interest anyway.
By the way, at first it was not clear to me that MaGe-ID numbers (for example 60141) are the last part of the entire locus tag (ARTHROv5_60141 for this example), and so this may be explained somewhere. Related to this, I suggest to include the underscore _ for the MaGe-IDs in each table (this has been done in Table 9, but not in others).
4.
The authors explain why they focused on protein-coding genes with a name (and thus, generally, with a predicted function). As a result, 85% of the regulated genes were disregarded (lines 28 & 1126). In my opinion, one of the aims of such an RNA-seq study is to potentially identify novel genes that may have a role in the response. Therefore it is a pity that unnamed genes were disregarded. I suggest to provide information for at least some of the unnamed genes that were found regulated, for example for genes highly induced with FC>5. For example, in Table S5 a sheet could be added with a list of all genes showing FC>5, and then for each predicted gene product relevant information can be mentioned, such as number of amino acids, description of conserved domain(s) found for the protein (giving at least a hint for a possible function), or lack of conserved domain, …
5.
Line 822: Even though msrP may have a mutation and msrQ is not found in L. indica PCC 8005, the MsrPQ system is described to function in the periplasm (in e.g. E. coli). Therefore, is it really possible to consider MrsA1 (cytoplasmic) and MsrPab as redundant?
6.
Line 332 (and elsewhere), the authors refer to the Methods section for the performed batch normalization, but as I do not see the words “batch normalization” in Materials and Methods, is this procedure really described?
Minor comments and typos:
Line 86: Therefore (not “Therefor”)
Figure 2: On my Mac computer, Fig 2 contains one or several grey lines when the manuscript pdf file is opened with Adobe Reader or with Preview, respectively. Also one or two dots are visible at upper left parts of some panels.
Line 278: not sanalysis but analysis
Line 286: I suppose it is not “i.e.” but “e.g.” (glnA is an example)
Line 444: I think it has to be “repressed” instead of “induced” (for kaiABC and cikA expression)
Line 457: the title of this section seems to have lost the symbol gamma.
Also the indicated number (3.2.4) of this section (and the next sections) appears incorrect because the preceding section is 3.2.1. For the same reason, the numbers when refering to sections need verification/correction (e.g. lines 559, 769, 875).
Similar for the section Conclusions (line 1061), which is given the number 5, but there is no section 4.
Line 519: space between gene and product (“geneproduct”)
Line 773: seemingly (not seeminly)
Line 885: restriction (not resitriction)
Line 956: Limnospira/Arthrospira (add “/”)
Author Response
Please see the attachment
2.13.0.0
Reviewer 2 Report
The present article written by Yadav et al. investigates the genetic responses of the two morphotypes of Limnospira indica exposed to gamma irradiation. Although the lack of functional information for the majority of L. indica PCC 8005 genes did not allow meaningful interpretation of their different metabolic and physiological responses to IR. The article is still informative for future studies in this field and proposes the information of specific genes requiring immediate attention in order to elucidate the difference to IR response. I have few comments before the publication of the manuscript:
1. The authors assume genetic differences are responsible for the difference in radioresistance mechanisms. However, for genetic analysis, the author does not use cells exposed to a cumulative dose of 5700 Gy assuming the highest cellular damage. It is not clear what kind of cellular damage do they indicate? The cells recovered well at 3200 Gy. Thus, the difference in the genetic expression pattern is expected to be low. The authors should justify their experimental scheme.
2. Although the authors investigated the radiation damage caused by gamma. It is worth mentioning that susceptibility to radiation damage (e.g. by UV) could also depend on the structure of RNA (ChemBioChem 2019, 20, 2609). For example, the duplex region of RNA is more protective to radiation damage irrespective of the possibility of pyrimidine dimer formation.
Author Response
Please see the attachment
2.13.0.0
Round 2
Reviewer 2 Report
The authors have addressed my concern. I recommend publication.